# TRAINING-FREE LLM-GENERATED TEXT DETECTION BY MINING TOKEN PROBABILITY SEQUENCES

**Yihuai Xu**[1,4], **Yongwei Wang**[1]*, **Yifei Bi**[1,2], **Huangsen Cao**[1], **Zhouhan Lin**[3], **Yu Zhao**[1], **Fei Wu**[1]
[1]Zhejiang University    [2]Georgia Institute of Technology
[3]Shanghai Jiao Tong University    [4]Zhejiang Gongshang University
{yongwei.wang, huangsen_cao, yuzhao, wufei}@zju.edu.cn
yihuai1024@outlook.com  lin.zhouhan@gmail.com  ybi48@gatech.edu

## ABSTRACT

Large language models (LLMs) have demonstrated remarkable capabilities in generating high-quality texts across diverse domains. However, the potential misuse of LLMs has raised significant concerns, underscoring the urgent need for reliable detection of LLM-generated texts. Conventional training-based detectors often struggle with generalization, particularly in cross-domain and cross-model scenarios. In contrast, training-free methods, which focus on inherent discrepancies through carefully designed statistical features, offer improved generalization and interpretability. Despite this, existing training-free detection methods typically rely on global text sequence statistics, neglecting the modeling of local discriminative features, thereby limiting their detection efficacy. In this work, we introduce a novel training-free detector, termed **Lastde**[1] that synergizes local and global statistics for enhanced detection. For the first time, we introduce time series analysis to LLM-generated text detection, capturing the temporal dynamics of token probability sequences. By integrating these local statistics with global ones, our detector reveals significant disparities between human and LLM-generated texts. We also propose an efficient alternative, **Lastde++** to enable real-time detection. Extensive experiments on six datasets involving cross-domain, cross-model, and cross-lingual detection scenarios, under both white-box and black-box settings, demonstrated that our method consistently achieves state-of-the-art performance. Furthermore, our approach exhibits greater robustness against paraphrasing attacks compared to existing baseline methods.

## 1 INTRODUCTION

Recent advancements in large language models (LLMs), such as GPT-4 (OpenAI, 2024b) and Gemma (Team et al., 2024), have significantly enhanced the text generation capabilities of machines. These models produce texts that are virtually indistinguishable from those written by humans, enabling broad applications across fields like journalism (Quinonez & Meij, 2024), education (M Alshater, 2022; Xiao et al., 2023), and medicine (Thirunavukarasu et al., 2023). However, the increasing sophistication of LLMs has also raised serious concerns about their potential misuse. Examples include the fabrication and dissemination of fake news (Opdahl et al., 2023; Fang et al., 2024), academic dishonesty in scientific writing (Else, 2023; Currie, 2023; Liang et al., 2024), and the risk of model collapse when trained on LLM-generated data (Shumailov et al., 2024; Wenger, 2024).

Prior arts in LLM-generated text detection can be broadly categorized as training-based and training-free methods. Training-based methods extract discriminative features from the texts and then input them into binary classifiers. These detectors require training on labeled datasets. Typical examples include RoBERTa-based fine-tuning (Guo et al., 2023), OpenAI Text Classifier (OpenAI, 2023), and GPTZero (Tian, 2023). In contrast, training-free methods utilize global statistical features of given texts such as likelihood and rank (Solaiman et al., 2019), which can be computed from LLMs during

---

*Corresponding author.

[1]The code and data are released at `https://github.com/TrustMedia-zju/Lastde_Detector`.

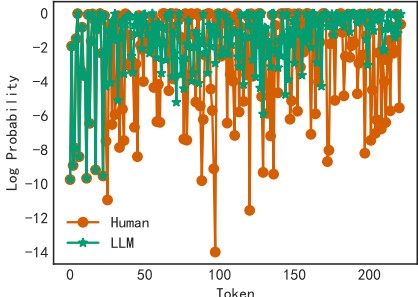
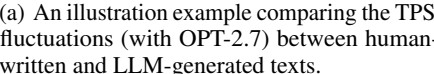
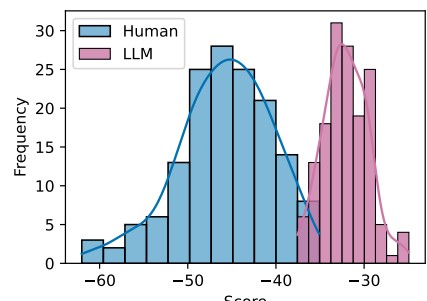

(a) An illustration example comparing the TPS fluctuations (with OPT-2.7) between human-written and LLM-generated texts.

(b) The proposed Lastde score distribution of 150 human and LLM-generated texts from the WritingPrompts dataset, using the OPT-2.7.

Figure 1: Comparison of TPS fluctuations and Lastde score distributions between human-written texts and LLM-generated texts, using the first 30 tokens of human texts as a prefix to continue writing.

the text inference. Typical training-free methods such as DetectGPT (Mitchell et al., 2023) and DNA-GPT (Yang et al., 2024) achieve detection by comparing or scoring these statistical features.

Training-based methods require significant time and effort in high-quality data collection, model training, etc. Also, they often face generalization issues, e.g. overfitting to in-domain data, particularly when detecting cross-domain texts Pu et al. (2023); Zhu et al. (2023). In contrast, training-free methods, which rely on scoring the analysis results of data statistics (often global statistics) or the data distribution, are more adept at handling cross-domain texts (Yu et al., 2024). However, existing training-free methods either incur high time and computational costs or perform poorly in scenarios involving cross-model detection and paraphrasing attacks. Additionally, most of these methods only evaluate the ***global*** statistical features (e.g. likelihood) of the text without harnessing the ***local*** statistical patterns (e.g. temporal dynamics) from token segments, significantly limiting the development and broader application of training-free methods.

To bridge the gap, this work aims to leverage local and global statistics for effective detection by mining discriminative patterns from the token probability sequence (TPS). Specifically, we first provide a novel perspective that views the TPS as a time series and harnesses time series analysis to extract local statistics. As illustrated in Figure 1(a), the TPS of human-written text fluctuates more abruptly than LLM-generated ones, showcasing distinct temporal dynamics. To quantify the dynamical complexity, we exploit the diversity entropy (Wang et al., 2020) that measures similarities among neighboring sliding-window segments of a TPS followed by conversion to discrete probability states for entropy calculation. We then aggregate the DEs (diversity entropy) from different time scales to establish our local statistics. Finally, we integrate the local statistical features with likelihood, a typical measure of global text features, to develop a novel detection method: **Lastde** (**L**ikelihood **a**nd **s**equential **t**oken probability **d**iversity **e**ntropy) and its variant **Lastde++**, with the latter being an enhanced version featuring fast-sampling for more efficient detection. As shown in Fig. 1(b), our method shows a clear distinguishability boundary between human and LLM-generated texts.

Extensive experiments across different datasets, models, and scenarios demonstrate that Lastde++ has a strong ability to distinguish between human and LLM-generated text. Notably, in the black-box setting, Lastde, relying on a single score, outperformed all existing single-score methods, even matching the detection performance of Fast-DetectGPT (Bao et al., 2024), a previous work that requires extensive sampling, thus our method establishes a new powerful benchmark.

Our main contributions can be summarized as follows:

1) We propose a novel and effective method for LLM-generated text detection, termed Lastde (and Lastde++), by mining token probability sequences from a time series analysis viewpoint.

2) We propose to leverage diversity entropy to capture the disparate temporal dynamics of a token probability sequence, extract the local statistics, and integrate them with the global statistics to achieve a robust detector.

3) Our method achieves state-of-the-art detection performance, surpassing advanced training-free baseline methods including DetectGPT, DNA-GPT, and Fast-DetectGPT with comparable or lower computational costs. Besides, it demonstrates superior robustness in handling complex detection scenarios, e.g. cross-domain, cross-model, cross-lingual tasks, and paraphrasing attacks.

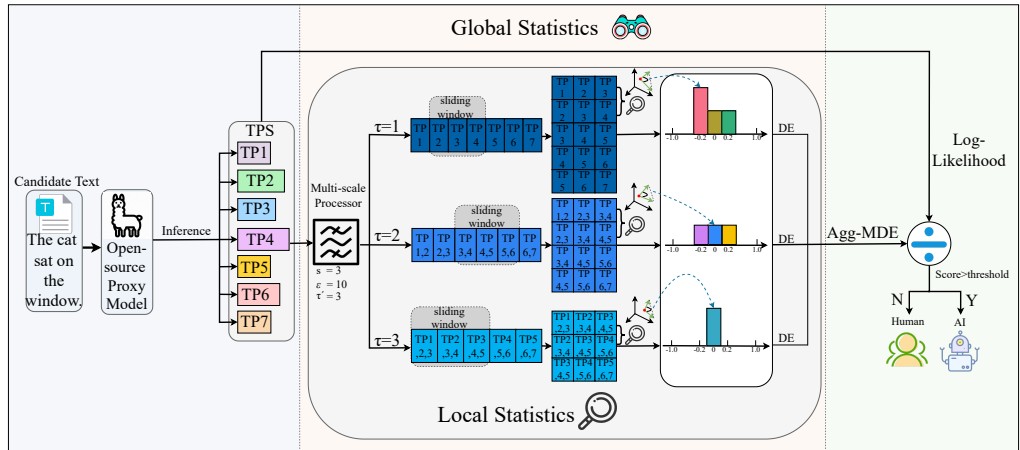

Figure 2: Overview of the *Lastde* detection framework. The example shows how a 7-token candidate text is fully detected with setting $s = 3, \varepsilon = 10, \tau' = 3$. First, a proxy model converts the text into a token (log) probability sequence (TPS). Then, both *global* and *local* statistics of the TPS are mined in parallel. The global statistics is straightforward. For local statistics, the TPS is mapped into 3 new sequences by a Multi-scale Processor (with $\tau' = 3$). Each scale factor $\tau_i \in \{1, 2, 3\}$ represents the arithmetic mean of $\tau_i$ adjacent elements from the original TPS. For each new sequence, a sliding window with a stride of 1 and width of 3 ($s = 3$) moves from left to right, forming a matrix by vertically stacking window segments' elements. The cosine similarity between adjacent rows (segments) is then calculated to generate a similarity sequence. We calculate the histogram of 10 ($\varepsilon = 10, [-1, 1]$) equally divided intervals based on the similarity sequence and derive the Diversity Entropy (DE) of the current sequence. Finally, we divide *Log-Likelihood* by aggregating all DEs (*Agg-MDE*) to derive the detection score, making a decision based on an appropriate threshold.

## 2 RELATED WORK

Detecting LLM-generated text has been a prominent area of research since the early days of large language models (Jawahar et al., 2020; Bakhtin et al., 2019; Crothers et al., 2023). Current studies predominantly focus on **training-based** and **training-free** detection methods.

Most training-based methods (Bhattacharjee et al., 2023; Li et al., 2023; Tian et al., 2023) leverage the **semantic features** of text (such as token embeddings) to train or fine-tune a classification model under the premise of labels for successful detection. Meanwhile, some approaches (Verma et al., 2023; Shi et al., 2024; Wang et al., 2023) incorporate probability information as part of the training features, which has also proven effective in detecting LLM-generated text in various scenarios.

However, research has shown that many existing training-based methods struggle with overfitting or generalizing to out-of-distribution data (Uchendu et al., 2020; Chakraborty et al., 2023). As a result, training-free methods have gained increasing attention aiming at identifying LLM-generated text across diverse domains and source models. Specifically, these methods focus on the **probabilistic features** of the text, scoring the text by constructing appropriate statistics, and ultimately making decisions based on a determined threshold. Representative methods include Likelihood, Rank (Solaiman et al., 2019), GLTR (Gehrmann et al., 2019), and DetectLRR (Su et al., 2023). DetectGPT Mitchell et al. (2023) pioneered the detection paradigm of contrasting perturbed texts with original text, subsequently leading to the development of DetectNPR (Su et al., 2023) and DNA-GPT (Yang et al., 2024). Yet we must acknowledge that these methods cannot achieve real-time or large-scale detection due to their excessive time consumption. Fast-DetectGPT (Bao et al., 2024) leverages fast-sampling technique to enhance the detection efficiency of DetectGPT by 340 times, thereby broadening the application prospects of training-free methods. However, its black-box detection performance still exhibits substantial potential for enhancement. More innovative detectors are continuously being conceived, please refer to (Mao et al., 2024; Hans et al., 2024).

We are confident that the constraints on the performance of existing training-free detectors in black-box detection arise from their failure to fully exploit or excavate the deeper layers of information

inherent in token probability sequence, particularly the local geometric information. To address this issue, we devised a detection statistic termed *Lastde*, which is employed to evaluate both human-written and LLM-generated texts. Our approach successfully establishes the connection between token probability sequence and traditional time series, achieving state-of-the-art performance among training-free methods and providing a novel perspective for the detection of LLM-generated texts.

## 3 METHODOLOGY

In this section, we first formulate the diversity entropy of a token probability sequence (TPS) to quantitatively measure its local dynamics. We then integrate local statistic features with the global ones (likelihood) to develop our Lastde detector, and further develop a fast-sampling alternative called Lastde++ for more efficient detection.

### 3.1 ANALYZING TPS WITH DIVERSITY ENTROPY

Diversity Entropy (DE) (Wang et al., 2020) is an entropy-based method for measuring the dynamic complexity of time series, originally applied to feature extraction in areas such as fault diagnosis, ECG, and signal processing. It is a process of first increasing the dimensionality and then reducing the dimensionality of a one-dimensional sequence. We restate and extend DE to make it suitable to detecting LLM-generated text (black-box) .

Given candidate text $t$ and a proxy model $M_\theta$, with the sliding window size $s \in \mathbb{N}^+$, the precision of the interval $\varepsilon \in \mathbb{N}^+$, and the number of scales $\tau' \in \mathbb{N}^+$. Input $t$ into $M_\theta$ for inference, we can obtain its (log) TPS

$$L(t) = \left(\log p_\theta(t_1|t_{<1}), \log p_\theta(t_2|t_{<2}), ..., \log p_\theta(t_n|t_{<n})\right), \tag{1}$$

where $t_i$ is the $i$-th token of $t$.

**Multiscale log-probability sequence.** Apply multiscale transformation to $L(t)$. Specifically, for $\tau = 1, 2, ..., \tau'$, let $L^{(\tau)}(t) = \left(\log p_\theta^{(\tau)}(j)\right)$ be the new TPS corresponding to the $\tau$-th scale transformation, where

$$\log p_\theta^{(\tau)}(j) = \frac{1}{\tau} \sum_{i=j}^{j+\tau-1} \log p_\theta(t_i|t_{<i}), \quad 1 \le j \le n - \tau + 1. \tag{2}$$

Note that, when $\tau = 1$, $L^{(1)}(t) = \left(\log p_\theta^{(1)}(j)\right) = (\log p_\theta(t_j|t_{<j})) = L(t)$, which is the original TPS of Equation 1. Different $\tau$ measures its dynamic characteristics at different scales, which is essentially an enhancement of the original data.

**Sliding-segmentation sequence.** Apply a sliding window of size $s$ and a step size of 1 to segment and rearrange $L^{(\tau)}(t), \tau = 1, 2, ..., \tau'$ in sequence, resulting in the following:

$$\begin{bmatrix} \log p_\theta^{(\tau)}(1) & \log p_\theta^{(\tau)}(2) & \cdots & \log p_\theta^{(\tau)}(s) \\ \log p_\theta^{(\tau)}(2) & \log p_\theta^{(\tau)}(3) & \cdots & \log p_\theta^{(\tau)}(1+s) \\ \vdots & \vdots & & \vdots \\ \log p_\theta^{(\tau)}(n-\tau-s+2) & \log p_\theta^{(\tau)}(n-\tau-s+1) & \cdots & \log p_\theta^{(\tau)}(n-\tau+1) \end{bmatrix}. \tag{3}$$

Denote the matrix in Equation 3 as

$$\tilde{L}^{(\tau)}(s) = \left[l_1^{(\tau)}(s), l_2^{(\tau)}(s), ..., l_{n-\tau-s+2}^{(\tau)}(s)\right]^T, \tag{4}$$

where $l_i^{(\tau)}(s) = \left(\log p_\theta^{(\tau)}(i), \log p_\theta^{(\tau)}(i+1), ..., \log p_\theta^{(\tau)}(i+s-1)\right)$ is the log-probability segment within window $i$ at the $\tau$-th scale, which is called an $s$-probability segment.

**Segment similarity.** For $\tilde{L}^{(\tau)}(s), \tau = 1, 2, ..., \tau'$ in Equation 4, compute the cosine similarity between adjacent $s$-probability segments, yielding the similarity sequence $D^{(\tau)}(s) = \left(d_1^{(\tau)}, d_2^{(\tau)}, ..., d_{n-\tau-s+1}^{(\tau)}\right)$, where

$$d_k^{(\tau)} = \text{CosineSimilarity}(l_k^{(\tau)}(s), l_{k+1}^{(\tau)}(s)), k = 1, 2, .., n - \tau - s + 1. \tag{5}$$

The geometric meaning of cosine similarity is the cosine of the angle between two vectors in space, with a range of values from [-1, 1]. Therefore, intuitively, the closer the similarity value of two adjacent $s$-probability segments is to 1, the smaller the deviation in direction, and the smaller the fluctuation in the local log-probability. Conversely, the larger the fluctuation in the local log-probability.

**Probability state sequence.** Divide the interval $[-1, 1]$ into $\varepsilon$ mutually exclusive and equally spaced sub-intervals, denoted as $I = (I_1, I_2, ..., I_\varepsilon)$, each of which is called a state. Therefore, we can easily obtain the statistical histogram $C^{(\tau)}(s) = (c_1^{(\tau)}, c_2^{(\tau)}, ..., c_\varepsilon^{(\tau)})$ of $D^{(\tau)}(s)$ on $I$, where $c_i^{(\tau)}$ is the number of elements in $D^{(\tau)}(s)$ that fall into the state $I_i$. Furthermore, we can calculate the probability state sequence

$$P^{(\tau)}(s) = (P_1^{(\tau)}, P_2^{(\tau)}, ..., P_\varepsilon^{(\tau)}), \tag{6}$$

where $P_i^{(\tau)} = \frac{c_i^{(\tau)}}{\sum_{k=1}^{\varepsilon} c_k^{(\tau)}}$. Obviously, the sum of the sequence of Equation 6 is equal to 1.

**Multiscale diversity entropy.** According to the DE formula of Equation 8, calculate the diversity entropy of the probability state sequence $P^{(\tau)}(s), \tau = 1, 2, ..., \tau'$ for each scale, and store them together to obtain the multiscale diversity entropy (MDE) sequence for the text $\boldsymbol{t}$, denoted as

$$\text{MDE}(\boldsymbol{t}, s, \varepsilon, \tau') = (\text{DE}(s, \varepsilon, 1), ..., \text{DE}(s, \varepsilon, \tau')), \tag{7}$$

where

$$\text{DE}(s, \varepsilon, \tau) = -\frac{1}{\ln \varepsilon} \sum_{i=1}^{\varepsilon} P_i^{(\tau)} \ln P_i^{(\tau)}, \quad P_i^{(\tau)} > 0. \tag{8}$$

According to (Shannon, 1948), the entropy value ranges from $[0, 1]$, and therefore, the DE value also ranges from $[0, 1]$. A DE value closer to 0 indicates that the TPS at the current scale has less fluctuation, while a value further from 0 indicates greater fluctuation.

## 3.2 LASTDE AND LASTDE++

**Lastde score.** Inspired by DetectLLM (Su et al., 2023), we first aggregated the MDE sequence of the text. Subsequently, we combined this with the Log-Likelihood of the text and defined the following **L**og-**L**ikelihood **a**nd **s**equential **t**oken probability multiscale **d**iversity **e**ntropy (Lastde) score

$$\text{Lastde}(\boldsymbol{t}, \theta) = \frac{\frac{1}{n} \sum_{i=1}^{n} \log p_\theta(t_i | t_{<i})}{\text{Agg}((\text{DE}(s, \varepsilon, 1), ..., \text{DE}(s, \varepsilon, \tau')))}, \tag{9}$$

where $\text{Agg} : \mathbb{R}^{\tau'} \rightarrow \mathbb{R}$ is an aggregation operator or function that summarizes the sequence's elements in a particular way, the rest of the notations are consistent with the previous definitions. Obviously, the numerator in Equation 9 is the Log-Likelihood value, which corresponds to the global statistical feature of TPS. Additionally, we refer to the result of the denominator in Equation 9 as the Agg-MDE value, which reflects the local statistical feature of the TPS. Unless explicitly stated, we use the standard deviation function (*Std*) as the aggregation function (*Agg*), additional aggregation functions and their detection results are located in Appendix D. Furthermore, for Lastde, the 3 hyperparameters are set to default values of $s = 3, \varepsilon = 10 \times n, \tau' = 5$, where $n$ is the number of tokens in the text. In summary, the detection process of Lastde can be outlined in three steps:

1. *Inference*: Retrieve the TPS of candidate text using a specific open-source proxy model.
2. *Computing statistics*: Simultaneously compute the Log-Likelihood and Agg-MDE statistics of the TPS.
3. *Scoring*: Divide the two statistics and make a decision based on the result.

A simple example illustrating the framework of our proposed algorithm can be seen in Figure 2.

We observed that the Lastde score distributions for the two types of texts are significantly different: the Lastde score for human-written texts is generally lower, while that for LLM-generated texts is higher, as shown in Figure 11. The ablation experiment in Appendix E also proves the rationality of applying the MDE algorithm to TPS. Therefore, to improve the detection performance of the Lastde method, we incorporate the fast-sampling technique (Bao et al., 2024) to modify the Lastde score, providing the final detection score of the text from the perspective of sampling discrepancy.

**Lastde++.** Specifically, for a given text $\boldsymbol{t}$, an available scoring model $M_\theta$, and a conditionally independent sampling model $M_{\tilde{\theta}}$, the sampling Lastde discrepancy is defined as

$$\mathbf{D}\left(\boldsymbol{t}, \theta, \tilde{\theta}\right) = \frac{\text{Lastde}(\boldsymbol{t}, \theta) - \tilde{\mu}}{\tilde{\sigma}}, \tag{10}$$

where $\text{Lastde}(\boldsymbol{t}, \theta)$ can be seen Equation 9,

$$\tilde{\mu} = \mathbb{E}_{\tilde{\boldsymbol{t}} \sim M_{\tilde{\theta}}(\tilde{\boldsymbol{t}}|\boldsymbol{t})} \left[\text{Lastde}(\tilde{\boldsymbol{t}}, \theta)\right] \quad \text{and} \quad \tilde{\sigma}^2 = \mathbb{E}_{\tilde{\boldsymbol{t}} \sim M_{\tilde{\theta}}(\tilde{\boldsymbol{t}}|\boldsymbol{t})} \left[(\text{Lastde}(\tilde{\boldsymbol{t}}, \theta) - \tilde{\mu})^2\right]. \tag{11}$$

where $\tilde{\mu}$ and $\tilde{\sigma}^2$ are the expectation and variance of the Lastde scores of the sampling samples $\tilde{\boldsymbol{t}}$ from $M_{\tilde{\theta}}$, respectively. Our default sampling number is 100, which is 1/100 of Fast-DetectGPT. In fact, when $\theta = \tilde{\theta}$, the sampling and scoring steps can be combined, and in this case, the detection task can be completed by only calling the model once. For Lastde++, the default settings are $s = 4, \varepsilon = 8 \times n, \tau' = 15$.

The discriminative power of Lastde++ in distinguishing human-written text and LLM-generated text is usually stronger than that of Lastde. As shown in Figure 12, the score distribution after sampling normalization is more favorable for distinguishing the two types of text compared to the Lastde score distribution. Appendix E presents a more detailed analysis of Lastde++.

## 4 EXPERIMENTS

### 4.1 SETTINGS

**Datasets.** The experiments conducted involved 6 distinct datasets, covering a range of languages and topics. Adhering to the setups of Fast-DetectGPT and DNA-GPT, we report the main detection results on 4 datasets: *XSum* (Narayan et al., 2018) (BBC News documents), *SQuAD* (Rajpurkar et al., 2016; 2018) (Wikipedia-based Q&A context), *WritingPrompts* (Fan et al., 2018) (for story generation),and *Reddit* ELI5 (Fan et al., 2019) (Q&A data restricted to the topics of biology, physics, chemistry, economics, law, and technique). To investigate the robustness of various detectors across different languages, we first conducted detection on *WMT16* (Bojar et al., 2016), a common dataset for English and German text translations. Subsequently, we gathered data on food, history, and economics from the popular Chinese Q&A platform *Zhihu*, extending detection to these Chinese datasets. Each dataset contained 150 human-written examples. For each human-written example, we used the first 30 tokens as a prompt to input into the source model and generate a LLM-generated continuation. More details on the dataset and prompt engineering are provided in the Appendix B.1.

**Source models & proxy model.** In order to comprehensively evaluate all the detectors, we utilized up to 18 source models for testing, including 15 famous open-source models with parameters ranging from 1.3B to 13B, as well as 3 of the latest and widely used closed-source models: `GPT-4-Turbo` (OpenAI, 2024b), `GPT-4o` (OpenAI, 2024a), and `Claude-3-haiku` (Antropic, 2024). Most of the open-source models were consistent with Fast-DetectGPT. Additionally, apart from mGPT (Shliazhko et al., 2022), Qwen1.5 (Bai et al., 2023), and Yi-1.5 (AI et al., 2024) being used for cross-lingual (German and Chinese) robustness detection, the residual 12 open-source models were employed to report the main detection results. Unless explicitly stated, all methods used GPT-J (Wang & Komatsuzaki, 2021) as the sole proxy model to execute the entire black-box detection process, encompassing rewriting (if necessary), sampling (if necessary), and scoring. Detailed information about all the source models can be found in the Appendix B.2.

**Baselines & metric.** We selected 8 representative training-free detection methods as baselines and categorized them into two main groups: *Sample-based* and *Distribution-based*. Both rely on statistics derived from the *Logits* tensor. The key difference is that the former directly utilizes the statistical values obtained as the final score for the text, while the latter generates a large number of contrast samples through specific methods (e.g., perturbation, rewriting, fast sampling) and then evaluates the overall distribution of these samples. Specifically, the first group includes Log-Likelihood (Solaiman et al., 2019), LogRank (Solaiman et al., 2019), Entropy (Gehrmann et al., 2019; Ippolito et al., 2019), and DetectLRR (Su et al., 2023). The second group consists of DetectGPT (Mitchell et al., 2023), DetectNPR (Su et al., 2023), DNA-GPT (Yang et al., 2024), and Fast-DetectGPT (Bao et al., 2024). Further details on the baselines can be found in the Appendix B.3. According to prior studies (Bao et al., 2024), the most commonly used evaluation metric is AUROC (area under the receiver operating characteristic curve). Therefore, we conform to this convention.

Table 1: Detection results for text generated by 12 source models under the white-box scenario. The AUROC values reported for each model are averaged across three datasets: *XSum*, *SQuAD*, and *WritingPrompts*. More detailed detection results are available in Table 6 in Appendix C.1. All methods use the source model for scoring. The "(*Diff*)" rows indicate the absolute improvement in AUROC of *Lastde* over *DetectLRR* for the first group of methods, and of *Lastde++* over *Fast-DetectGPT* for the second group. "value" denotes the second-best AUROC. Implementation details for all baselines can be found in the Appendix B.3.

| Methods/Models | GPT-2 | Neo-2.7 | OPT-2.7 | GPT-J | Llama-13 | Llama2-13 | Llama3-8 | OPT-13 | BLOOM-7.1 | Falcon-7 | Gemma-7 | Phi2-2.7 | Avg. |
|---|---|---|---|---|---|---|---|---|---|---|---|---|---|
| *Sample-based Methods* | | | | | | | | | | | | | |
| Likelihood | 91.65 | 89.40 | 88.08 | 84.95 | 63.66 | 65.36 | 98.35 | 84.45 | 88.00 | 76.78 | 70.14 | 89.67 | 82.54 |
| LogRank | 94.31 | 92.87 | 90.99 | 88.68 | 68.87 | 70.27 | 99.04 | 87.74 | 92.42 | 81.32 | 74.81 | 92.13 | 86.12 |
| Entropy | 52.15 | 51.72 | 50.46 | 54.31 | 64.18 | 61.05 | 23.30 | 54.30 | 62.67 | 59.33 | 66.47 | 44.09 | 53.67 |
| DetectLRR | 96.67 | 96.07 | 93.13 | 92.24 | 81.40 | 80.89 | 98.94 | 91.03 | 96.35 | 87.45 | 81.36 | 94.10 | 90.80 |
| **Lastde** | **98.41** | **98.64** | **98.15** | **97.24** | **88.98** | **88.40** | **99.71** | **96.47** | **99.35** | **95.49** | **91.85** | **96.99** | **95.89** |
| *(Diff)* | 1.74 | 2.57 | 5.02 | 5.00 | 8.58 | 7.51 | 0.77 | 5.44 | 3.00 | 8.04 | 10.5 | 2.89 | 5.09 |
| *Distribution-based Methods* | | | | | | | | | | | | | |
| DetectGPT | 93.43 | 90.40 | 90.36 | 83.82 | 63.78 | 65.39 | 70.13 | 85.05 | 89.28 | 77.98 | 68.96 | 89.55 | 80.68 |
| DetectNPR | 95.77 | 94.77 | 93.24 | 88.86 | 68.60 | 69.83 | 95.55 | 89.78 | 94.95 | 83.06 | 74.74 | 93.06 | 86.85 |
| DNA-GPT | 89.92 | 86.80 | 86.79 | 82.21 | 62.28 | 64.46 | 98.07 | 82.51 | 86.74 | 74.04 | 63.63 | 88.00 | 80.45 |
| Fast-DetectGPT | 99.57 | 99.49 | 98.78 | 98.95 | 93.45 | 93.34 | 99.91 | 98.07 | 99.53 | 97.74 | 96.90 | 98.10 | 97.82 |
| **Lastde++** | **99.76** | **99.87** | **99.46** | **99.52** | **96.58** | **96.67** | 99.82 | **98.77** | **99.84** | **98.76** | **98.40** | **98.76** | **98.85** |
| *(Diff)* | 0.19 | 0.38 | 0.68 | 0.57 | 3.13 | 3.33 | -0.09 | 0.70 | 0.31 | 1.02 | 1.50 | 0.66 | 1.03 |

Table 2: Detection results for text generated by 11 open-source models and 1 closed-source model (GPT-4-Turbo) under the black-box scenario. The AUROC values reported for each model are averaged across three datasets: *XSum*, *WritingPrompts*, and *Reddit*. The meanings of "(*Diff*)" and "value" are the same as in Table 1. More detailed detection results and results for the remaining 2 closed-source models are provided in the Appendix C.2 and D.

| Methods/Models | GPT-2 | Neo-2.7 | OPT-2.7 | Llama-13 | Llama2-13 | Llama3-8 | OPT-13 | BLOOM-7.1 | Falcon-7 | Gemma-7 | Phi2-2.7 | GPT-4-Turbo | Avg. |
|---|---|---|---|---|---|---|---|---|---|---|---|---|---|
| *Sample-based Methods* | | | | | | | | | | | | | |
| Likelihood | 65.88 | 67.09 | 67.40 | 65.75 | 68.61 | 99.60 | 68.80 | 61.80 | 67.42 | 69.90 | 73.93 | 79.69 | 71.32 |
| LogRank | 70.38 | 71.17 | 72.35 | 70.28 | 72.67 | **99.69** | 73.01 | 67.51 | 71.66 | 72.17 | 77.99 | 79.24 | 74.84 |
| Entropy | 61.48 | 58.65 | 54.55 | 49.14 | 45.18 | 14.43 | 53.09 | 60.84 | 50.55 | 48.01 | 46.58 | 35.09 | 48.13 |
| DetectLRR | 79.30 | 79.19 | 81.25 | 78.51 | 78.94 | 97.35 | 80.27 | 79.57 | 79.87 | 73.47 | 83.79 | 73.85 | 80.45 |
| **Lastde** | **89.17** | **90.24** | **89.70** | **80.71** | **79.90** | 99.67 | **90.01** | **88.94** | **84.36** | **79.61** | **88.32** | 81.33 | **86.38** |
| *(Diff)* | 9.87 | 11.1 | 8.45 | 2.20 | 0.96 | 2.32 | 9.74 | 9.37 | 4.49 | 6.14 | 4.53 | 7.48 | 6.38 |
| *Distribution-based Methods* | | | | | | | | | | | | | |
| DetectGPT | 67.56 | 69.28 | 72.03 | 66.12 | 67.96 | 82.90 | 73.89 | 61.83 | 68.69 | 66.55 | 72.76 | 81.73 | 70.94 |
| DetectNPR | 68.07 | 68.41 | 73.06 | 67.83 | 70.60 | 96.75 | 75.13 | 63.00 | 70.42 | 65.72 | 74.08 | 79.94 | 72.75 |
| DNA-GPT | 64.15 | 62.63 | 63.64 | 60.77 | 66.71 | 99.47 | 65.75 | 62.01 | 65.08 | 62.59 | 72.02 | 70.75 | 67.97 |
| Fast-DetectGPT | 89.82 | 88.75 | 86.52 | 77.58 | 77.62 | 99.43 | 86.16 | 84.55 | 81.42 | 81.49 | 86.67 | 88.18 | 85.68 |
| **Lastde++** | **94.93** | **95.28** | **94.13** | **85.00** | **85.80** | 99.03 | **93.37** | **92.22** | **89.49** | **87.58** | **92.67** | **88.21** | **91.47** |
| *(Diff)* | 5.11 | 6.53 | 7.62 | 7.42 | 8.18 | -0.40 | 7.21 | 7.67 | 8.07 | 6.09 | 6.00 | 0.03 | 5.80 |

## 4.2 MAIN RESULTS

We introduce Lastde as a new sample-based method to compare with the first group and Lastde++ as its enhanced version to compare with the second group, following the default settings described in Sec. 3.2. In the white-box scenario, we use the 12 open-source models mentioned earlier to generate LLM-generated text and perform detection concurrently. In the black-box scenario, since GPT-J has already been used as the proxy model, it is not suitable to use it as the source model again, so we deploy GPT-4-Turbo to fill that gap.

**Overall results.** Table 1 and 2 present the detection results under white-box and black-box scenarios. Overall, Lastde and Lastde++, which require only 100 samples, consistently outperform DetectLRR and Fast-DetectGPT in both scenarios. Specifically, in the white-box scenario, Lastde achieves an average AUROC of 95.89% across the 12 source models, significantly surpassing methods of the same type and approaching the performance levels of Fast-DetectGPT. In the black-box scenario, Lastde not only outperforms Fast-DetectGPT in detecting most open-source models but its enhanced version, Lastde++, further improves the average AUROC by 5.80% . In summary, Lastde++ significantly improves detection performance for open-source models in black-box scenarios compared to Fast-DetectGPT and also achieves the best performance on closed-source models, making it a superior training-free detection method. Given that black-box scenarios are more common in practice and that open-source models still have a wide range of applications similar to closed-source models due to factors such as API call costs, Lastde++ is poised to become a more accurate cross-source LLM-generated texts detector and a powerful benchmark. For detailed detection results, please refer to the Table 6 in the Appendix C.1 and the Table 7 in the C.2.

**Hyperparameter analysis.** In Figure 3, we explored the impact of Lastde's three hyperparameters $(s, \varepsilon, \tau')$ on detection results across four specific datasets. Firstly, a larger window size results in

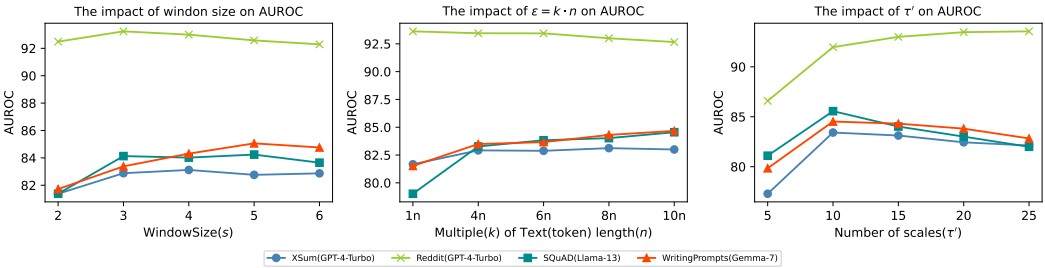

Figure 3: Sensitivity analysis results for three types of hyperparameters in *Lastde++*. The legend denotes (*Dataset*, *Source Model*). Through preliminary analysis and referring to the parameter tuning experiments of the original paper of the MDE algorithm, we explore the following ranges: $s \in \{2, 3, 4, 5, 6\}$, $\varepsilon \in \{n, 4n, 6n, 8n, 10n\}$, $\tau' \in \{5, 10, 15, 20, 25\}$. In each experiment, we adjust only one type of hyperparameters while keeping the other two types fixed at their default settings.

fewer $s$-probability segments, while if the size is too small, the reflection of the variations will be too monotonous. Both situations are not conducive to capturing the authentic information of the sequence. This is confirmed by Figure 3 (left), where a size of 3 or 4 optimally balances detection performance across the four datasets. Secondly, a larger $\varepsilon$ results in finer granularity when partitioning the $[-1, 1]$ range, which is generally advantageous. To adaptively vary $\varepsilon$, it is set as a multiple ($k$) of the number of tokens ($n$). As shown in Figure 3 (middle), detection performance improves with increasing $k$ on most datasets, converging at 8x or 10x, except for the Reddit (GPT-4-Turbo) dataset. Finally, increasing the scale factor from 5 to 10 improves detection across all four datasets. However, beyond 10, performance continues to improve on the Reddit (GPT-4-Turbo) dataset but plateaus or slightly declines on others, likely because a huge scale factor may not align well with the token count.

### 4.3 ROBUSTNESS ANALYSIS

**Samples number.** According to findings from DetectLLM (Su et al., 2023), increasing the number of perturbed samples enhances detection accuracy. Our experiments on contrast samples for all distribution-based methods reveal that more contrast samples improve detection performance, with DetectGPT shows the most significant gains (see Figure 4). Notably, Lastde++ achieves superior performance with just 10 samples, surpassing Fast-DetectGPT with 100 samples and outperforming other methods by a considerable margin. This underscores Lastde++'s strong competitiveness even with a limited number of contrast samples.

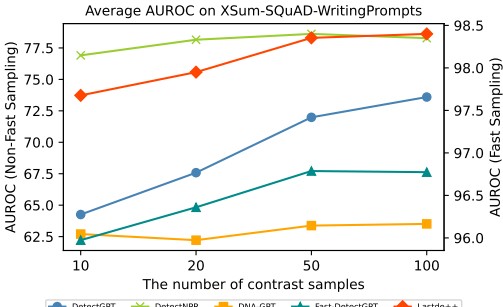

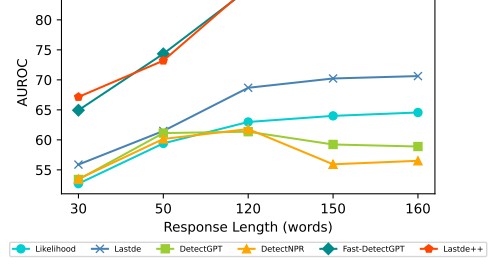

Figure 4: Distribution-based methods' robustness to contrast sample numbers. The *right* y-axis represents the AUROC of *fast* sampling methods, while the *left* y-axis represents *non-fast* sampling methods (see Appendix F.1). Perform white-box detection using *Gemma-7* as the source model.

Figure 5: Detection results of 6 detection methods on 5 response lengths. Specifically, the 3 hyperparameters of Lastde++ and Lastde were set to $s = 3, \varepsilon = 1 \cdot n, \tau' = 10$ to adapt to shorter text. The settings for the other methods were kept at their default settings.

**Response lengths.** Prior studies (Verma et al., 2023; Mao et al., 2024) have demonstrated that shorter texts are generally less conducive to detection. Therefore, we evaluated detection performance across varying response lengths (number of words), focusing primarily on XSum generated by GPT-4-Turbo

due to source response length constraints. We truncated responses to $\{30, 50, 120, 150, 160\}$ words, revealing in Figure 5 that most detection methods' performance improves with longer responses, consistent with prior research findings. The shorter response length inherently restricts the number of scales ($\tau'$), thereby diminishing the capacity of Lastde and Lastde++ to capture local information in the initial stages. When the length exceeds 120, Lastde and Lastde++ clearly outperform other comparable methods, underscoring their sustained competitiveness.

**Decoding strategies.** We evaluated the impact of different decoding strategies on detection, including top-$p$, top-$k$, and temperature sampling, each controlling the diversity of generated text from different angles. We conducted comparisons on Xsum, SQuAD, and WritingPrompts. The results presented in Figure 6 indicate that across all three strategies, Lastde++ consistently exhibits superior robustness among the three source models, while Lastde significantly outperforms sample-based methods such as DetectLRR, achieving detection performance comparable to Fast-DetectGPT.

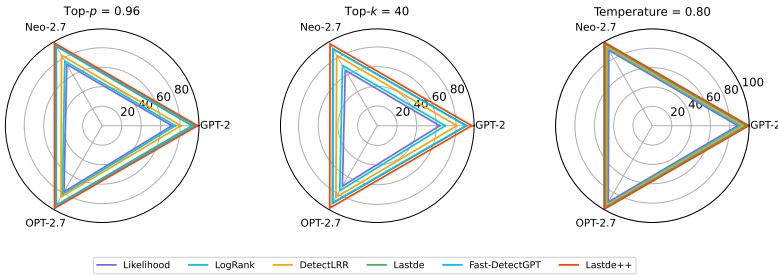

Figure 6: The impact of the decoding strategy. The average AUROC values across 3 datasets are provided for each Detection method (detailed results can be found in Table 15 in Appendix F.3). The source models were *GPT-2*, *Neo-2.7*, and *OPT-2.7*, with the decoding strategy hyperparameters set to top-$p = 0.96$, top-$k = 40$, and temperature=0.8. All experiments were conducted in a black-box setting, with other model implementation details kept at their default settings.

**Paraphrasing attack.** Previous studies (Krishna et al., 2024; Sadasivan et al., 2023) show that paraphrasing attacks can effectively evade detection by methods like watermarking, GPTZero, DetectGPT, and OpenAI's text classifier. Following Bao (Bao et al., 2024), we used T5-Paraphraser[2] to perform paraphrasing attacks on texts generated by source models. As shown in Table 3, both methods exhibited performance drops in detecting paraphrased texts, in both black-box and white-box scenarios. However, Lastde++ consistently maintained a significant advantage, with smaller performance declines compared to Fast-DetectGPT across most scenarios and datasets. Notably, in detecting paraphrased Reddit (GPT-4-Turbo), Lastde++ surpassed Fast-DetectGPT, indicating superior robustness (see Appendix F.2).

Table 3: AUROC for *Fast-DetectGPT* and *Lastde++* before and after paraphrasing attacks, with average values across three source models in both white-box (*Llama-13*, *OPT-13*, *GPT-J*) and black-box (*Llama-13*, *OPT-13*, *GPT-4-Turbo*) scenarios, and the decrease in AUROC values indicated in parentheses. Detailed results are shown in the Appendix F.2.

| Methods/Datasets | XSum(Original) | XSum(Paraphrased) | WritingPrompts(Original) | WritingPrompts(Paraphrased) | Reddit(Original) | Reddit(Paraphrased) |
|---|---|---|---|---|---|---|
| *White-box Settings* | | | | | | |
| Fast-DetectGPT | 96.04 | 85.48 ($\downarrow$ 10.6) | 98.99 | 96.29 ($\downarrow$ 2.70) | 98.08 | 94.60 ($\downarrow$ 3.48) |
| **Lastde++** | **97.75** | **89.27 ($\downarrow$ 8.48)** | **99.57** | **97.84 ($\downarrow$ 1.73)** | **99.10** | **97.00 ($\downarrow$ 2.10)** |
| *Black-box Settings* | | | | | | |
| Fast-DetectGPT | 76.62 | 67.17 ($\downarrow$ 9.45) | 89.26 | 83.82 ($\downarrow$ 5.44) | 86.04 | 83.04 ($\downarrow$ **3.00**) |
| **Lastde++** | **82.57** | **73.43 ($\downarrow$ 9.14)** | **92.74** | **88.54 ($\downarrow$ 4.20)** | **91.26** | **87.90 ($\downarrow$ 3.36)** |

---

[2] https://huggingface.co/Vamsi/T5_Paraphrase_Paws

**Non-English scenarios.** The latest research (Liang et al., 2023) shows that detectors exhibit bias against non-English writers. Therefore, we conducted experiments on datasets in three different languages, and the results confirmed that Lastde++ remains effective (see Figure 7). The Lastde++ outperforms in detecting texts across all three languages. Specifically, Lastde++ surpasses Fast-DetectGPT by an average AUROC margin of 4.56% (WMT16-German) and 5.24% (WMT16-English) in both scenarios. On our newly introduced Chinese dataset, which includes three topics, both methods perform well, but Lastde++ demonstrates a slight advantage. Therefore, Lastde++ is better suited for cross-lingual detection scenarios compared to Fast-DetectGPT.

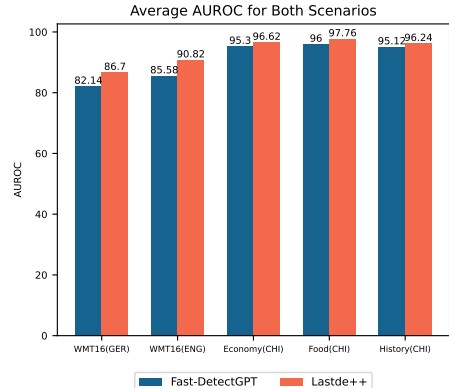

Figure 7: Detection performance comparison across *English*, *German*, and *Chinese* texts. The source models in the white-box scenario are mGPT (English and German) and Qwen1.5-7 (Chinese), while the source models in the black-box scenario remain unchanged and the proxy models are GPT-J (English and German) and Yi1.5-6 (Chinese).

**Selection of proxy models.** Bao (Bao et al., 2024) discussed the impact of proxy sampling models on detection performance in white-box scenarios. Our experiments further reveal that the choice of proxy model in black-box scenarios significantly influences detection performance. We evaluated five training-free detection methods across five different source and proxy model pairs, as illustrated in Figure 8. The results indicate that Lastde matches in most or even surpasses Fast-DetectGPT pairings, while Lastde++ outperforms Fast-DetectGPT in all five pairings. Notably, detection performance was poor across all five methods with the closed-source GPT-4-Turbo and the proxy model Llama3-8. Despite this, our proposed Lastde and Lastde++ consistently outperformed Fast-DetectGPT and other baseline methods, further emphasizing that the choice of proxy model greatly influences detection performance.

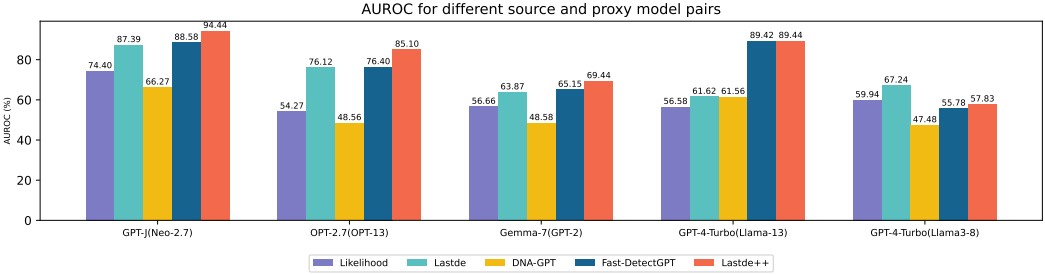

Figure 8: The performance of 5 zero-shot detection methods under 5 different combinations of source models and proxy models is presented. The X-axis represents the 5 different combinations, with the proxy model in parentheses and the source model outside the parentheses. The source dataset is *XSum*, and the other settings are kept at their default values.

## 5 CONCLUSION

In this work, we propose a novel training-free method for detecting LLM-generated text, termed Lastde and Lastde++, by mining token probability sequences (TPS). By observing significant differences in the temporal dynamics of TPS between human-written and LLM-generated texts, we leveraged diverse entropy from time series analysis to develop local statistical features, integrating them with global statistics to create an effective and robust detector. Our approach demonstrates efficacy in both black-box and white-box settings while maintaining comparable or lower computational costs. Furthermore, it outperforms existing training-free detectors in complex scenarios, including paraphrasing attacks and cross-lingual tasks.

## ACKNOWLEDGMENTS

This work was supported in part by Shanghai Pujiang Program (No. 23PJ1412100), Yangtze River Delta Community of Sci-Tech Innovation (Grant No. 2023CSJZN0301), and National Natural Science Foundation of China (No. U24A20326, 62441605).

## ETHICS STATEMENT

Lastde and Lastde++ are novel LLM-generated text detectors capable of real-time detection, achieving new advancements in complex scenarios such as cross-domain and cross-lingual. However, as we mentioned in the paper, they still produce imperfect detection results when faced with sophisticated challenges such as paraphrasing attacks and the arbitrary replacement of proxy models. Therefore, we strongly recommend that users exercise prudence when interpreting the detection results. The results should be regarded as supplementary references to support, rather than replace, human judgment in critical decision-making processes.

## REPRODUCIBILITY STATEMENT

All experimental projects in this paper are reproducible. We provide details on the generation of all datasets and the implementation details of all baselines in Appendix B. Additionally, we provide the source addresses for the third-party datasets in Appendix D.

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

## A    TASK AND SCENARIO DEFINITION FOR TRAINING-FREE DETECTION

**Task Definition.** Current research mainly defines the problem of LLM-generated text detection in two ways. According to DetectGPT (Mitchell et al., 2023), training-free detection can be defined as: given a specific autoregressive language model $M$ and a candidate text $\boldsymbol{t} = (t_1, t_2, ..., t_n)$ containing $n$ tokens, determine whether $\boldsymbol{t}$ was generated by $M$ without any training. Under this setup, even if the judgment is "no", the candidate text may still belong to another language model $Q(\neq M)$. Therefore, we can only conclude that $\boldsymbol{t}$ is not generated by $M$, rather than by humans. This definition is obviously more suitable for more precise detection situations such as *tracing* the source of text. In practical detection scenarios, we often seek a more straightforward conclusion. So in this paper, we follow the definition provided by Fast-DetectGPT (Bao et al., 2024) for the training-free detection problem, which is: given a text $\boldsymbol{t}$, determine whether $\boldsymbol{t}$ is LLM-generated without any machine training. This definition allows us to use any available language model (proxy model) for inference and judgment.

**Scenarios definition.** Suppose $\boldsymbol{t}$ is a LLM-generated text, we can classify the detection scenarios into white-box and black-box settings (Gehrmann et al., 2019) based on whether we can obtain the probabilities $p_\phi(t_i|t_{<i})$ for each token $t_i$ from source model $M_\phi$ that generated $\boldsymbol{t}$, where $\phi$ is the parameters of $M$. In the white-box setting, the range of detection results is limited to the source model and humans. We input $\boldsymbol{t}$ into $M_\phi$ for inference, and the returned result contains actual statistical information about $\boldsymbol{t}$, such as the probability of each token. In a black-box setting, detection becomes more challenging because we cannot know in advance whether $\boldsymbol{t}$ was generated by $M$. As a result, it is impossible to use $M_\phi$ to obtain the actual statistical information about $\boldsymbol{t}$. Fortunately, there are many open-source models (such as LLaMA Series (Touvron et al., 2023a;b; AI@Meta, 2024)) with varying parameter scales that we can leverage. These models share a similar underlying architecture, with the common goal of learning the general patterns of human text creation. Based on this consideration, a compromise approach for black-box scenario detection is to select a suitable open-source model (not necessarily the source model itself) as a proxy model, and indirectly realize detection through analyzing the proxy model's inference results on $\boldsymbol{t}$.

Indeed, black-box detection is more common in real-world scenarios. With the release of increasingly powerful open-source and closed-source models, developing a universal detector that can operate across different sources and domains to improve the accuracy of detecting LLM-generated texts in black-box scenarios is becoming an urgent task. However, white-box detection also plays an irreplaceable role in tasks such as text attribution (He et al., 2024), where it is sometimes crucial to precisely identify the model that generated the text, a task that is no simpler than black-box detection.

## B    DETAILS ABOUT THE SETTINGS AND BASELINES

### B.1    DATASETS AND PROMPTS

*XSum*, *SQuAD*, and *WritingPrompts* were primarily used for white-box detection, while *XSum*, *WritingPrompts*, and *Reddit* were primarily used for black-box detection. For each dataset, we filtered out samples with text length less than 150 words, and randomly sampled 150 of these as human-written examples. For each human-written example, we used the first 30 tokens as a prompt to generate a continuation from the source model. For source models that require system prompts or API calls, including OpenAI's `gpt-4-turbo-2024-04-09`, `gpt-4o-2024-05-13`, and Anthropic's `claude-3-haiku-20240307`, we provide the details of prompt engineering in Table 4.

Table 4: Examples of prompts used in different datasets for Black-box detection.

| Datasets | Prompts |
|---|---|
| Xsum | **System:** You are a professional News writer. |
| | **User:** Please write an article with about 200 words starting exactly with: *Prefix* |
| Writing Prompts | **System:** You are a professional Fiction writer. |
| | **User:** Please write a story with about 200 words starting exactly with: *Prefix* |
| Reddit | **System:** You are a helpful assistant that answers the question provided. |
| | **User:** Please continue the answer in 200 words starting exactly with: *Prefix* |

Thus, each complete dataset contained 150 negative (human-written) and 150 positive (LLM-generated) samples, with roughly equivalent text length and the beginning portion between each pair. Finally, We have excluded the commonly used *PubMedQA* dataset (Jin et al., 2019) from our experiments because the length of most samples in this dataset is significantly shorter compared to the other datasets. This discrepancy in sample length could compromise the fairness of the testing.

## B.2 ALL LARGE LANGUAGE MODELS USED

All the source models used are listed in Table 5, which includes both open-source and closed-source models. Our experimental setup consists of two RTX 3090 GPUs ($2\times24$GB). Due to GPU memory limitations, we did not conduct experiments on models with parameter sizes greater than 20B, such as NeoX (Black et al., 2022), and instead replaced it with Falcon-7, Gemma-7, and Phi2-2.7 to cover as many large language model products from tech companies as possible. Except for Neo-2.7, GPT-2, OPT-2.7, Phi2-2.7, and mGPT, which use full-precision (float32), the rest of the open-source models use half-precision (float16).

Table 5: Details of the source models that is used to produce machine-generated text.

| Model | Model File/Service | Parameters |
|---|---|---|
| mGPT (Shliazhko et al., 2022) | ai-forever/mGPT | 1.3B |
| GPT-2 (Radford et al., 2019) | openai-community/gpt2-xl | 1.5B |
| OPT-2.7 (Zhang et al., 2022) | facebook/opt-2.7b | 2.7B |
| Neo-2.7 (Black et al., 2021) | EleutherAI/gpt-neo-2.7B | 2.7B |
| Phi2-2.7 (Gunasekar et al., 2023) | microsoft/phi-2 | 2.7B |
| GPT-J (Wang & Komatsuzaki, 2021) | EleutherAI/gpt-j-6B | 6B |
| Yi1.5-6 (AI et al., 2024) | 01-ai/Yi-1.5-6B | 6B |
| Falcon-7 (Penedo et al., 2023) | tiiuae/falcon-7b | 7B |
| Gemma-7 (Team et al., 2024) | google/gemma-7b | 7B |
| Qwen1.5-7 (Bai et al., 2023) | Qwen/Qwen1.5-7B | 7B |
| BLOOM-7.1 (Workshop et al., 2022) | bigscience/bloom-7b1 | 7.1B |
| Llama3-8 AI@Meta (2024) | meta-llama/Meta-Llama-3-8B | 8B |
| OPT-13 (Zhang et al., 2022) | facebook/opt-13b | 13B |
| Llama-13 (Touvron et al., 2023a) | huggyllama/llama-13b | 13B |
| Llama2-13 (Touvron et al., 2023b) | TheBloke/Llama-2-13B-fp16 | 13B |
| GPT-4-Turbo (OpenAI, 2024b) | OpenAI | NA |
| GPT-4o (OpenAI, 2024a) | OpenAI | NA |
| Claude-3-haiku (Antropic, 2024) | Anthropic | NA |

It is worth noting that the default setting of Fast-DetectGPT uses GPT-J as the conditional independent sampling model and Neo-2.7 as the scoring model. However, we did not follow this setting and instead unified the sampling model and scoring model as GPT-J. On the one hand, Fast-DetectGPT (our main improved method) achieves the best performance compared to itself in our dataset. On the other hand, using a single model not only saves GPU memory but also makes the detection process more unified.

## B.3 DETAILS OF BASELINES

The implementation details of the 8 baselines in this article are as follows:

**Log-Likelihood** (Solaiman et al., 2019). The average log probability of all tokens in the candidate text is used as the metric.

**Log-Rank** (Solaiman et al., 2019). The average log rank of all tokens in the candidate text, sorted in descending order by probability, is used as the metric.

**Entropy** (Gehrmann et al., 2019; Ippolito et al., 2019). The average entropy of each token is calculated based on the probability distribution over the vocabulary.

**DetectLRR** (Su et al., 2023). The ratio of log-likelihood to log-rank is used as the average metric.

The first group of methods aggregates the probability information of each token in the text by constructing appropriate statistics, which are used as the final detection score. By default, we use GPT-J as the scoring model in black-box scenario.

**DetectGPT** (Mitchell et al., 2023). Perturbation likelihood discrepancy is used as the metric. We maintain the default settings from the original paper, using T5-3B as the perturbation model with 100 perturbations.

**DetectNPR** (Su et al., 2023). Normalized log-rank perturbation is used as the metric, with all other settings consistent with DetectGPT.

**DNA-GPT** (Yang et al., 2024). Contrast samples are generated using a cut-off and rewrite method, with log-likelihood as the metric. The default settings are a truncation rate of $\tau = 0.5$ and 10 rewrites.

**Fast-DetectGPT** (Bao et al., 2024). Sampling discrepancy is used as the metric. To ensure a fair comparison, we chose both the sampling model and the scoring model as GPT-J, instead of the original setting where the sampling model was GPT-J and the scoring model was Neo-2.7. This modification results in better detection performance on our dataset. The number of samples remains at 10,000.

The second group of methods requires perturbing or sampling the original samples to obtain a certain number of contrast samples, then calculating the statistical discrepancy between the contrast samples and the original samples as the final detection score. In the black box scenario, we use GPT-J as the sampling, rewriting, and scoring model by default. Non-sampling methods (e.g., DetectNPR and DNA-GPT) generally require multiple model calls, making detection time longer. Therefore, sampling-based methods are more practical in comparison.

## C   DETAILS OF DETECTION RESULTS

### C.1   DETAIL OF MAIN RESULTS IN THE WHITE-BOX SCENARIO

The main results of the white-box scenario are shown in Table 6. First, the basic version of the Lastde method achieved the best detection performance among the 5 sample-based training-free methods, with an average AUROC 5.09% higher than the previously best-performing DetectLRR method. Notably, it led by approximately 8% in the well-known LLaMA series and Gemma, making it a strong detection statistics. Second, among the 5 distribution-based training-free methods, the enhanced Lastde++ method achieved the best detection performance with only 100 samples, significantly fewer than Fast-DetectGPT. This demonstrates that our method has a notable advantage in the white-box scenario, providing insights for more detailed detection tasks such as model attribution.

### C.2   DETAIL OF MAIN RESULTS IN THE BLACK-BOX SCENARIO

Although the detection performance of all methods across the 12 source models showed a significant decline compared to the white-box scenario, as shown in Table 7, Lastde and Lastde++ still outperformed other methods of the same type. Lastde++ maintained a considerable leading advantage across all open-source and closed-source models. Notably, the two fast-sampling methods, Fast-DetectGPT and Lastde++, performed worse on Llama3-8 compared to their base statistics, Likelihood and Lastde. A possible reason for this is that the detection performance may have already saturated, and the sampling process introduced additional randomness.

Additionally, in the black-box scenario, DetectLRR showed a noticeable performance gap in detecting GPT-4-Turbo compared to Likelihood and LogRank, whereas this was not observed in the other open-source models. This observation is consistent with the conclusions of Fast-DetectGPT (Bao et al., 2024), and we provide further analysis of this phenomenon in Appendix D.

Table 6: The details in Table 1 include the main results of 8 baseline methods as well as Lastde and Lastde++ on the white-box detection task across the XSum, SQuAD, and WritingPrompts datasets. DNA-GPT generates rewritten samples using the source model, while Fast-DetectGPT and Lastde++ utilize sampling from the source model. Other implementation settings are described in Appendix B.3.

| Methods/Model | GPT-2 | Neo-2.7 | OPT-2.7 | GPT-J | Llama-13 | Llama2-13 | Llama3-8 | OPT-13 | BLOOM-7.1 | Falcon-7 | Gemma-7 | Phi2-2.7 | Avg. |
|---|---|---|---|---|---|---|---|---|---|---|---|---|---|
| | | | | | | *XSum* | | | | | | | |
| Likelihood | 86.89 | 86.68 | 81.46 | 81.47 | 61.66 | 63.08 | 99.43 | 77.15 | 85.31 | 68.60 | 68.04 | 88.68 | 79.04 |
| LogRank | 90.00 | 90.59 | 84.64 | 84.99 | 67.07 | 68.75 | 99.86 | 80.99 | 90.61 | 73.71 | 73.16 | 91.29 | 82.97 |
| Entropy | 56.19 | 59.82 | 54.33 | 59.80 | 66.32 | 62.99 | 34.77 | 57.34 | 69.58 | 67.09 | 68.73 | 56.24 | 59.43 |
| DetectLRR | 92.49 | 92.44 | 85.57 | 86.69 | 78.95 | 76.96 | 99.01 | 84.42 | 95.11 | 80.13 | 81.86 | 91.73 | 87.11 |
| **Lastde** | 95.79 | 97.44 | 95.63 | 96.52 | 89.92 | 85.72 | 100.0 | 92.32 | 99.12 | 92.68 | 92.28 | 95.82 | 94.44 |
| DetectGPT | 89.52 | 89.41 | 83.33 | 81.24 | 68.81 | 67.55 | 69.27 | 76.95 | 86.99 | 75.70 | 68.08 | 88.97 | 78.82 |
| DetectNPR | 90.98 | 92.31 | 86.46 | 86.15 | 69.49 | 70.94 | 98.52 | 80.01 | 93.67 | 77.09 | 73.85 | 92.64 | 84.34 |
| DNA-GPT | 83.78 | 80.30 | 77.08 | 75.35 | 59.25 | 58.83 | 98.40 | 71.32 | 80.82 | 60.15 | 59.36 | 83.64 | 74.02 |
| Fast-DetectGPT | 98.92 | 98.97 | 96.98 | 98.12 | 94.80 | 92.63 | **99.99** | 95.19 | 99.22 | 96.20 | 96.40 | 98.08 | 97.12 |
| **Lastde++** | 99.36 | 99.72 | 98.46 | 99.04 | 97.27 | 96.11 | 99.71 | 96.94 | 99.64 | 97.91 | 97.98 | 98.97 | 98.43 |
| | | | | | | *SQuAD* | | | | | | | |
| Likelihood | 92.05 | 86.77 | 88.84 | 80.47 | 46.56 | 48.36 | 95.65 | 84.04 | 85.08 | 75.60 | 72.79 | 83.21 | 78.29 |
| LogRank | 95.14 | 91.48 | 92.39 | 85.84 | 51.84 | 53.55 | 97.25 | 87.92 | 90.55 | 81.20 | 76.77 | 87.36 | 82.61 |
| Entropy | 59.86 | 57.88 | 52.77 | 61.14 | 70.47 | 71.44 | 30.76 | 61.02 | 67.35 | 63.11 | 63.16 | 50.94 | 59.16 |
| DetectLRR | 97.97 | 96.92 | 95.83 | 92.58 | 71.65 | 73.27 | 98.93 | 92.16 | 95.33 | 89.29 | 80.72 | 92.64 | 89.77 |
| **Lastde** | 99.62 | 98.87 | 99.01 | 96.39 | 81.99 | 82.61 | 99.16 | 98.08 | 99.21 | 95.92 | 96.88 | 96.88 | 94.88 |
| DetectGPT | 93.73 | 86.26 | 90.15 | 77.83 | 40.96 | 47.16 | 54.71 | 81.76 | 86.81 | 70.52 | 71.46 | 81.66 | 73.58 |
| DetectNPR | 97.48 | 94.13 | 94.68 | 84.40 | 48.30 | 50.32 | 90.66 | 91.15 | 93.60 | 80.99 | 76.74 | 88.09 | 82.54 |
| DNA-GPT | 93.74 | 88.78 | 89.96 | 83.33 | 50.65 | 54.83 | 96.56 | 85.43 | 88.22 | 79.94 | 67.68 | 86.44 | 80.46 |
| Fast-DetectGPT | 99.91 | 99.72 | 99.69 | 99.51 | 87.20 | 89.35 | **99.91** | 99.62 | 99.64 | 98.75 | 96.79 | 99.28 | 97.45 |
| **Lastde++** | 99.96 | 99.93 | 99.95 | 99.78 | 92.90 | 94.50 | 99.84 | 99.97 | 99.93 | 99.42 | 98.80 | 99.30 | 98.69 |
| | | | | | | *WritingPrompts* | | | | | | | |
| Likelihood | 96.02 | 94.76 | 93.93 | 92.92 | 82.76 | 84.64 | 99.98 | 92.17 | 93.62 | 86.13 | 69.58 | 97.13 | 90.30 |
| LogRank | 97.78 | 96.55 | 95.94 | 95.22 | 87.69 | 88.52 | 100.00 | 94.30 | 96.09 | 89.04 | 74.49 | 97.73 | 92.78 |
| Entropy | 40.39 | 37.45 | 44.28 | 41.98 | 55.74 | 48.71 | 4.36 | 44.53 | 51.07 | 47.78 | 67.52 | 25.10 | 42.41 |
| DetectLRR | 99.54 | 98.85 | 97.99 | 97.44 | 93.60 | 92.45 | 99.88 | 96.50 | 98.60 | 92.94 | 81.49 | 97.93 | 95.52 |
| **Lastde** | 99.73 | 99.60 | 99.80 | 98.82 | 98.05 | 96.86 | 99.98 | 99.01 | 99.71 | 97.88 | 92.47 | 98.26 | 98.35 |
| DetectGPT | 97.04 | 95.52 | 97.61 | 92.40 | 81.56 | 81.45 | 86.42 | 96.45 | 94.04 | 87.73 | 67.35 | 98.02 | 89.63 |
| DetectNPR | 98.85 | 97.88 | 98.57 | 96.04 | 88.00 | 88.23 | 97.48 | 98.19 | 97.59 | 91.11 | 73.62 | **98.45** | 93.67 |
| DNA-GPT | 92.25 | 91.32 | 93.34 | 87.95 | 76.95 | 79.72 | 99.26 | 90.77 | 91.19 | 82.04 | 63.85 | 93.91 | 86.88 |
| Fast-DetectGPT | 99.89 | 99.78 | 99.68 | 99.22 | 98.35 | 98.04 | 99.91 | **99.40** | 99.74 | 98.27 | 97.50 | 96.94 | 98.89 |
| **Lastde++** | 99.96 | 99.95 | 99.96 | 99.75 | 99.56 | 99.41 | 99.92 | 99.40 | 99.95 | 98.95 | 98.42 | 98.02 | 99.44 |

Table 7: The details in Table 2 include the main results of 8 baseline methods, as well as Lastde and Lastde++, on the black-box detection task across the XSum, WritingPrompts, and Reddit datasets. All baselines use the default settings described in Appendix B.3 for the black-box scenario.

| Methods/Model | GPT-2 | Neo-2.7 | OPT-2.7 | Llama-13 | Llama2-13 | Llama3-8 | OPT-13 | BlOOM-7.1 | Falcon-7 | Gemma-7 | Phi2-2.7 | GPT-4-Turbo | Avg. |
|---|---|---|---|---|---|---|---|---|---|---|---|---|---|
| | | | | | | *XSum* | | | | | | | |
| Likelihood | 54.02 | 48.45 | 63.20 | 54.22 | 54.32 | 98.91 | 68.84 | 39.49 | 54.02 | 65.43 | 54.61 | 60.44 | 59.66 |
| LogRank | 58.35 | 53.52 | 66.96 | 60.09 | 59.46 | 99.12 | 72.30 | 47.25 | 59.82 | 68.14 | 61.64 | 61.52 | 64.01 |
| Entropy | 65.46 | 69.85 | 56.85 | 53.78 | 51.57 | 29.08 | 53.21 | 73.31 | 60.20 | 51.65 | 64.27 | 61.24 | 57.54 |
| DetectLRR | 67.38 | 66.48 | 71.76 | 71.82 | 68.17 | 96.25 | 74.99 | 66.08 | 71.25 | 70.24 | 74.45 | 61.71 | 71.72 |
| **Lastde** | 79.98 | 82.17 | 83.47 | 72.30 | 67.01 | 99.32 | 86.95 | 77.46 | 75.63 | 76.23 | 79.44 | 64.16 | 78.68 |
| DetectGPT | 58.86 | 55.97 | 64.31 | 56.28 | 54.98 | 75.66 | 65.44 | 46.87 | 55.83 | 62.28 | 54.22 | 67.35 | 59.78 |
| DetectNPR | 55.64 | 50.72 | 63.58 | 56.50 | 55.65 | 96.36 | 66.72 | 44.54 | 56.33 | 60.37 | 55.15 | 62.94 | 60.37 |
| DNA-GPT | 53.17 | 44.47 | 55.40 | 52.92 | 54.63 | 99.04 | 59.26 | 48.67 | 54.63 | 55.03 | 54.28 | 57.32 | 57.40 |
| Fast-DetectGPT | 80.80 | 77.22 | 82.14 | 64.79 | 62.81 | 99.20 | 84.27 | 69.45 | 72.17 | 76.11 | 75.70 | 80.79 | 77.17 |
| **Lastde++** | 88.78 | 89.09 | 91.32 | 72.84 | 73.00 | 98.80 | 91.76 | 82.20 | 82.79 | 83.15 | 85.41 | 83.12 | 85.19 |
| | | | | | | *WritingPrompts* | | | | | | | |
| Likelihood | 78.06 | 82.36 | 77.08 | 76.45 | 78.99 | 99.88 | 77.43 | 76.75 | 78.60 | 66.20 | 86.19 | 81.48 | 79.96 |
| LogRank | 82.04 | 85.72 | 81.98 | 79.75 | 81.80 | 99.96 | 81.38 | 81.38 | 81.48 | 68.03 | 88.67 | 79.03 | 82.60 |
| Entropy | 54.04 | 48.93 | 48.69 | 45.06 | 39.68 | 6.150 | 44.92 | 52.43 | 44.11 | 49.17 | 36.99 | 35.56 | 42.14 |
| DetectLRR | 88.61 | 90.52 | 90.58 | 85.27 | 85.96 | 99.82 | 88.00 | 89.92 | 86.40 | 67.34 | 90.49 | 66.75 | 85.72 |
| **Lastde** | 95.27 | 96.82 | 96.09 | 88.02 | 87.32 | 99.97 | 94.12 | 96.16 | 92.95 | 73.58 | 93.68 | 83.09 | 91.42 |
| DetectGPT | 77.05 | 78.47 | 83.30 | 74.46 | 77.79 | 90.14 | 84.65 | 70.29 | 80.28 | 62.14 | 84.86 | **94.21** | 79.80 |
| DetectNPR | 79.36 | 81.29 | 83.48 | 77.03 | 81.13 | 96.48 | 86.16 | 73.04 | 81.41 | 58.75 | 85.64 | 90.78 | 81.21 |
| DNA-GPT | 74.27 | 75.82 | 73.55 | 68.35 | 75.52 | **99.56** | 74.20 | 71.53 | 71.74 | 58.84 | 82.55 | 72.82 | 74.90 |
| Fast-DetectGPT | 95.86 | 96.07 | 92.03 | 88.26 | 86.44 | 99.56 | 89.64 | 92.75 | 90.04 | 76.53 | 93.63 | 88.50 | 90.89 |
| **Lastde++** | 98.99 | 98.93 | 97.19 | 94.46 | 93.05 | 98.84 | 95.25 | 97.39 | 95.84 | 84.31 | 96.92 | 88.50 | 94.97 |
| | | | | | | *Reddit* | | | | | | | |
| Likelihood | 65.55 | 70.45 | 61.91 | 66.58 | 72.53 | 100.00 | 60.14 | 69.16 | 69.65 | 78.08 | 81.01 | 97.15 | 74.35 |
| LogRank | 70.76 | 74.27 | 68.12 | 71.00 | 76.76 | 100.00 | 65.35 | 73.89 | 73.68 | 80.34 | 83.66 | **97.16** | 77.92 |
| Entropy | 64.95 | 57.16 | 58.11 | 48.59 | 44.29 | 8.050 | 61.14 | 56.77 | 47.33 | 43.21 | 38.48 | 8.480 | 44.71 |
| DetectLRR | 81.91 | 80.57 | 81.41 | 78.44 | 82.68 | 96.98 | 77.83 | 82.71 | 81.95 | 82.84 | 86.44 | 93.10 | 83.90 |
| **Lastde** | 92.28 | 91.71 | 89.53 | 81.80 | 85.34 | 99.73 | 88.98 | 93.20 | 84.49 | 89.27 | 91.84 | 96.74 | 90.41 |
| DetectGPT | 66.77 | 73.41 | 69.18 | 67.63 | 71.11 | 82.91 | 71.59 | 68.32 | 69.97 | 75.22 | 79.19 | 83.63 | 73.24 |
| DetectNPR | 69.20 | 73.23 | 72.13 | 69.97 | 75.03 | 97.40 | 72.51 | 71.41 | 73.51 | 78.03 | 81.44 | 86.10 | 76.66 |
| DNA-GPT | 65.01 | 67.59 | 61.96 | 61.04 | 69.99 | 99.82 | 63.78 | 65.84 | 68.88 | 73.90 | 79.24 | 82.12 | 71.60 |
| Fast-DetectGPT | 92.79 | 92.97 | 85.39 | 79.68 | 83.60 | 99.53 | 84.57 | 91.46 | 82.04 | 91.22 | 90.68 | **93.87** | 88.98 |
| **Lastde++** | 97.01 | 97.82 | 93.90 | 87.70 | 91.35 | 99.45 | 93.09 | 97.06 | 89.84 | 95.28 | 95.67 | 93.00 | 94.26 |

# D ADDITIONAL RESULTS ON DIFFERENT AGGREGATION FUNCTIONS AND MORE CLOSED-SOURCE MODEL DETECTION

We conducted additional tests of Lastde and Lastde++ on more closed-source models, and we also reported the impact of different aggregation functions on these two detection methods. As illustrated in Figure 8, Lastde(Std) performs comparably to the classic Likelihood on GPT-4o, while maintaining a significant advantage over other methods in the same group on the other two advanced closed-source

Table 8: The AUROC value of the closed-source model. When using Lastde or Lastde++, we considered five aggregation functions. Specifically, *2-norm* refers to calculating the 2-norm of the sequence; *Range* refers to calculating the range of the sequence; *Std* refers to calculating the standard deviation of the sequence (which is our default setting); *ExpRange* and *ExpStd* refer to taking the exponential of the range and standard deviation of the sequence, respectively. It is worth noting that we adjust the 3 hyperparameters of Lastde to $s = 3, \varepsilon = 1, \tau' = 15$ to enhance detection performance. Lastde++ and the other methods maintain their default settings.

| Methods SourceModels → ↓ Datasets → | GPT-4-Turbo | | | | GPT-4o | | | | Claude-3-haiku | | | |
|---|---|---|---|---|---|---|---|---|---|---|---|---|
| | XSum | WritingPrompts | Reddit | Avg. | XSum | WritingPrompts | Reddit | Avg. | XSum | WritingPrompts | Reddit | Avg. |
| Likelihood | 60.44 | 81.48 | 97.15 | 79.69 | **75.42** | 84.90 | **97.74** | **86.02** | 96.84 | 98.38 | 99.92 | 98.38 |
| LogRank | 61.52 | 79.03 | **97.16** | 79.24 | 73.85 | 82.32 | **97.74** | 84.64 | 97.09 | 98.71 | **99.96** | 98.59 |
| Entropy | 61.24 | 35.56 | 08.48 | 35.09 | 47.50 | 31.60 | 09.74 | 29.61 | 38.90 | 17.69 | 06.56 | 21.05 |
| DetectLRR | 61.71 | 66.75 | 93.10 | 73.85 | 62.87 | 69.06 | 93.75 | 75.23 | 95.78 | 97.96 | 99.56 | 97.77 |
| **Lastde(Std)** | **64.16** | **83.09** | 96.74 | **81.33** | 73.87 | **86.20** | **97.74** | 85.94 | **97.44** | **99.40** | 99.92 | **98.92** |
| *Aggregation function* | | | | | | | | | | | | |
| Fast-DetectGPT | 80.79 | **89.88** | 93.87 | 88.18 | 86.87 | 93.77 | 97.93 | 92.86 | 99.93 | 99.99 | 99.96 | 99.96 |
| Lastde++(2-Norm) | 76.91 | 87.39 | 93.61 | 85.97 | 85.74 | 92.96 | 97.52 | 92.07 | 99.95 | 99.99 | 99.96 | 99.97 |
| Lastde++(Range) | 82.67 | 86.37 | 91.72 | 86.92 | 85.96 | 91.34 | 96.57 | 91.29 | 99.84 | 99.99 | 99.95 | 99.93 |
| Lastde++(Std) | **83.12** | 88.50 | 93.00 | 88.21 | 86.47 | 93.41 | 96.98 | 92.29 | 99.92 | 99.96 | 99.89 | 99.92 |
| **Lastde++(ExpRange)** | 82.40 | 89.02 | 93.70 | 88.37 | **87.42** | 93.60 | 97.77 | 92.93 | 99.96 | **100.00** | **99.97** | **99.98** |
| **Lastde++(ExpStd)** | 81.55 | 89.81 | **93.99** | **88.45** | 87.24 | **94.24** | **97.94** | **93.14** | **99.97** | 99.99 | 99.95 | 99.97 |

models. Furthermore, appropriate aggregation functions are beneficial for improving the performance of Lastde++.

Intuitively, using ExpStd as the aggregation function seems to be a better choice than Std. However, when using it to detect open-source models, we found that although Lastde++(ExpStd) also achieves state-of-the-art results, its advantage is far less pronounced than that of Lastde++(Std), which is in stark contrast to the results on the aforementioned 3 closed-source models. Possible reasons include model size, hyperparameter settings, choice of proxy models, and the applicability of rapid sampling techniques. Given the larger role of open-source models in the current development of LLMs, we continue to use Std as the default aggregation function and report most of the experimental results accordingly, highlighting the significant performance improvements of Lastde++ over previous detection methods.

Next, in order to further enhance the persuasiveness of our approachs, we conducted extended experiments on several third-party datasets. The details are as follows:

**Including more datasets.** We conducted experiments on multiple datasets recently released by ReMoDetect[3] (Lee et al., 2024) and Fast-DetectGPT[4] (Bao et al., 2024). These datasets cover four domains: XSum, SQuAD, WritingPrompts, PubMedQA, and six *chat/instruct* type or *large-parameter* ($\geq$ 20B) source models: NeoX-20B, Llama3-70B, GPT-3.5-Turbo, GPT-4, GPT-4-Turbo, Claude3. For Fast-DetectGPT, we only use the dataset generated by NeoX-20B.

**Including more baselines.** We have added another two strong baselines: Binoculars (Hans et al., 2024), GPTZero (commercial, 2024-11-11 base version) (Tian, 2023). For Binoculars, we adopted its default settings from the original paper. For GPTZero, we first replicated the detection results provided in the ReMoDetect paper, then implemented our own detection on Fast-DetectGPT's dataset related to NeoX-20B.

**Combining with plug-and-play detector.** We explored the feasibility of integrating our method with recent plug-and-play detectors (plugins), e.g. TOCSIN (Ma & Wang, 2024). Specifically, we combine TOCSIN with Lastde, Lastde++ and other detectors (except Binoculars, GPTZero), and report the detection results (AUROC values) on the above multiple datasets.

**Unified hyperparameters.** Here, we unify the hyperparameters of Lastde and Lastde++ to $s = 3, \varepsilon = 10, \tau' = 5$. Note that this is not the best hyperparameter used by Lastde++ in our paper and our dataset, but the following results still prove that it can achieve SOTA performance. When testing the three datasets related to NeoX-20B, we set the aggregation function of Lastde and Lastde++ to $Agg = Std$ (default setting), and set it to $Agg = ExpStd$ when testing the other source models.

---

[3] https://github.com/hyunseoklee-ai/ReMoDetect/tree/main/exp/data
[4] https://github.com/baoguangsheng/fast-detect-gpt/tree/main/exp_main/data

We conducted experiments under the above settings and divided the results into two parts: *before* and *after* combining with TOCSIN. The results are as follows: It can be observed that even without running Lastde++ under the optimal hyperparameter settings, our method still achieves SOTA performance. This conclusion remains valid even when compared to two newly introduced baselines: Binoculars (using two proxy models with larger parameters, 7B > 6B) and the latest version of GPTZero. Additionally, Lastde demonstrates the best performance among sample-based methods. Most importantly, the experimental results in Table 9 were all obtained on third-party datasets, further demonstrating the reliability of our method.

Table 9: Detection results (AUROC) **before** combining with TOCSIN. It should be noted that Binoculars uses *Falcon-7B* as observer, *Falcon-7B-Instruct* as performer, and uses the judgment threshold under Low-FPR given by the author: $0.8536432310785527$.

| Models | Domains | Likelihood | LogRank | DetectLRR | Fast-DetectGPT | Binoculars | GPTZero | **Lastde** | **Lastde++** |
|---|---|---|---|---|---|---|---|---|---|
| NeoX-20B | XSum | 68.1 | 69.9 | 70.6 | 84.7 | 54.3 | 60.1 | 78.8 | 88.4 |
| | SQuAD | 64.8 | 69.3 | 77.6 | 85.9 | 56.6 | 52.6 | 86.2 | 91.8 |
| | WritingPrompts | 87.7 | 89.7 | 90.7 | 96.7 | 66.1 | 72.4 | 94.7 | 97.4 |
| Llama3-70B | XSum | 96.9 | 97.4 | 94.9 | 99.9 | 98.7 | 100 | 97.2 | 99.9 |
| | WritingPrompts | 98.2 | 97.9 | 93.8 | 99.9 | 100 | 99.8 | 98.2 | 99.9 |
| | PubMedQA | 84.8 | 83.5 | 71.1 | 90.5 | 88.7 | 90.1 | 85.6 | 90.7 |
| GPT-4-Turbo | XSum | 87.8 | 89.1 | 87.2 | 98.3 | 94.9 | 100 | 88.9 | 98.5 |
| | WritingPrompts | 98.2 | 98.0 | 94.4 | 99.9 | 99.3 | 100 | 98.2 | 99.8 |
| | PubMedQA | 85.8 | 85.1 | 74.5 | 87.8 | 90.0 | 87.2 | 86.4 | 87.8 |
| GPT-4 | XSum | 73.7 | 73.7 | 69.8 | 91.6 | 78.7 | 98.2 | 74.1 | 91.6 |
| | WritingPrompts | 87.6 | 85.5 | 73.4 | 97.6 | 90.7 | 82.6 | 87.3 | 97.4 |
| | PubMedQA | 79.6 | 79.0 | 69.3 | 83.7 | 86.7 | 84.8 | 80.4 | 83.8 |
| GPT-3.5-Turbo | XSum | 93.8 | 94.1 | 90.9 | 99.2 | 99.7 | 99.5 | 94.2 | 99.1 |
| | WritingPrompts | 98.1 | 97.7 | 93.0 | 99.7 | 99.3 | 92.9 | 98.2 | 99.6 |
| | PubMedQA | 87.2 | 86.6 | 75.8 | 90.4 | 93.0 | 88.0 | 88.1 | 90.4 |
| Claude-3 | XSum | 91.8 | 92.6 | 89.8 | 96.4 | 94.0 | 99.9 | 92.4 | 96.4 |
| | WritingPrompts | 97.0 | 96.4 | 88.8 | 96.1 | 96.0 | 99.1 | 97.0 | 96.0 |
| | PubMedQA | 85.5 | 84.9 | 74.8 | 88.0 | 88.0 | 88.0 | 86.2 | 87.9 |
| **Avg.** | | 87.0 | 87.2 | 82.2 | 93.7 | 87.5 | 88.6 | 89.6 | **94.2** |

Table 10: The detection results (AUROC) **after** combining with TOCSIN. When using the TOCSIN module, the two hyperparameters $n$ (the number of copies created for each input) and $\rho$ (the proportion of tokens to be deleted in each copy) are set to $10$ and $15\%$ respectively.

| Models | Domains | Likelihood | LogRank | DetectLRR | Fast-DetectGPT | **Lastde** | **Lastde++** |
|---|---|---|---|---|---|---|---|
| NeoX-20B | XSum | 97.7 | 97.6 | 99.0 | 89.4 | 98.9 | 96.8 |
| | SQuAD | 88.8 | 88.9 | 94.8 | 88.6 | 95.7 | 95.9 |
| | WritingPrompts | 98.1 | 98.2 | 98.0 | 97.9 | 99.3 | 99.2 |
| Llama3-70B | XSum | 99.2 | 99.4 | 97.2 | 99.9 | 99.3 | 99.9 |
| | WritingPrompts | 99.7 | 99.6 | 96.8 | 100 | 99.7 | 100 |
| | PubMedQA | 84.5 | 83.2 | 68.9 | 90.5 | 85.2 | 90.7 |
| GPT-4-Turbo | XSum | 85.2 | 87.7 | 73.5 | 98.3 | 86.3 | 98.3 |
| | WritingPrompts | 98.0 | 98.0 | 88.8 | 99.9 | 98.0 | 99.8 |
| | PubMedQA | 85.4 | 84.7 | 70.8 | 87.8 | 86.1 | 87.7 |
| GPT-4 | XSum | 96.8 | 96.5 | 96.3 | 93.8 | 97.0 | 93.7 |
| | WritingPrompts | 94.2 | 93.2 | 81.7 | 97.8 | 94.1 | 97.8 |
| | PubMedQA | 78.9 | 78.4 | 65.4 | 83.7 | 79.8 | 83.9 |
| GPT-3.5-Turbo | XSum | 99.8 | 99.8 | 99.6 | 99.4 | 99.8 | 99.4 |
| | WritingPrompts | 99.3 | 99.3 | 99.3 | 99.7 | 99.3 | 99.8 |
| | PubMedQA | 86.4 | 85.7 | 71.9 | 90.2 | 87.3 | 90.2 |
| Claude-3 | XSum | 88.3 | 91.2 | 74.6 | 96.4 | 89.2 | 96.7 |
| | WritingPrompts | 97.5 | 97.6 | 90.5 | 96.6 | 97.5 | 96.5 |
| | PubMedQA | 85.6 | 84.8 | 72.3 | 88.0 | 86.1 | 88.5 |
| **Avg.** | | 92.4 | 92.4 | 85.5 | 94.3 | 93.3 | **95.3** |

All detection methods show performance improvements when combined with the lightweight plug-and-play detection module TOCSIN. Notably, Lastde demonstrates a more significant gain, with an improvement of 1.1%, compared to 0.6% for Fast-DetectGPT. With the increasing dataset and strong baselines, we believe that Lastde and Lastde++ are efficient detectors.

# E ANALYSIS OF THE APPLICATIONS OF MDE AND ABLATION EXPERIMENTS OF LASTDE

In this section, we present further application cases of MDE Wang et al. (2020) and Lastde on token probability sequence across various types of texts, which substantiate the validity of our prior observations and hypotheses.

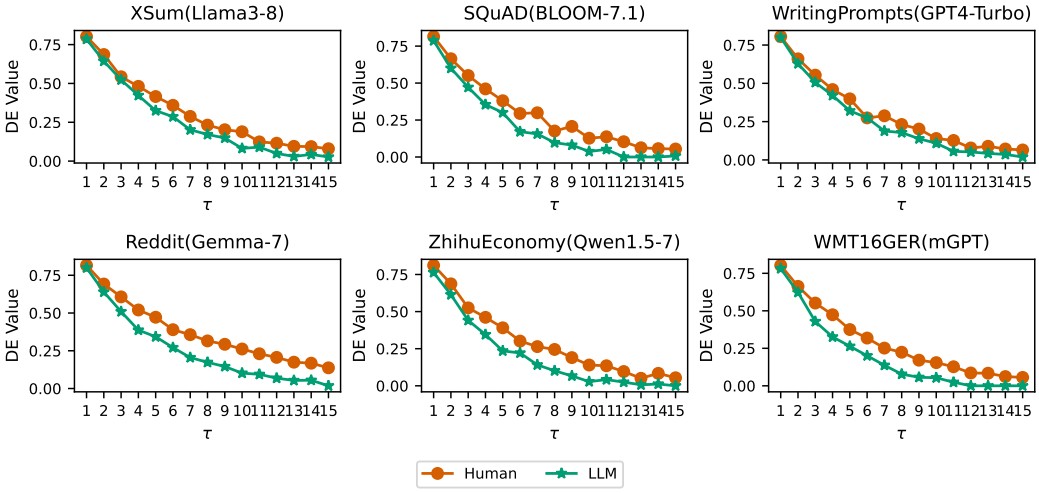

Figure 9: The comparison results of the MDE curves for human-LLM texts. The subtitles indicate the dataset or source model associated with each example. In addition to the scale number $\tau'$, the other two parameters are set as follows: $s = 3$ and $\varepsilon = 1$. The example WritingPrompts (GPT-4-Turbo) is conducted in a black-box scenario, while the remaining examples are examined in white-box contexts.

**Properties of MDE transformation.** Based on our observations from Figure 1(a), we found that under the same prefix, the dynamic variations in the token probability sequences for human-written subsequent texts and LLM-generated response texts differ. The token probabilities of LLM-generated responses are more compact, whereas human responses exhibit the opposite trend. This is attributable to the fact that LLMs are trained with the objective of minimizing perplexity. Therefore, according to the principles of MDE, the DE values corresponding to human texts are theoretically expected to be larger. We performed MDE transformations on 6 pairs of examples, setting the scale number $\tau'$ to 15 for each text in every pair. Subsequently, we calculated the DE values for each scale, resulting in the MDE sequence comparison shown in Figure 9.

It is evident that the curve representing humans consistently lies above the curve representing LLMs, and this conclusion holds across various languages, scenarios, and models. Interestingly, for both humans and LLMs, there is an overall downward trend in the DE values of the transformed TPS as the scale increases. This suggests that after multiple filtering iterations, the token probability sequences gradually approach a periodic or deterministic state. In summary, these results illustrate that the token probability sequences of humans and LLMs can be differentiated through MDE sequences, thereby validating the reasonableness of our hypothesis.

Table 11: Detailed results of the ablation experiments under the black-box setting. Using both Agg-MDE and Log-Likelihood corresponds to the standard Lastde. The rest of the settings are default.

| Scoring Types | | Xsum | | | | WritingPrompts | | | | Reddit | | | |
|---|---|---|---|---|---|---|---|---|---|---|---|---|---|
| Agg-MDE | Log-Likelihood | GPT-2 | Neo-2.7 | OPT-2.7 | Avg. | GPT-2 | Neo-2.7 | OPT-2.7 | Avg. | GPT-2 | Neo-2.7 | OPT-2.7 | Avg. |
| ✘ | ✔ | 54.02 | 48.45 | 63.20 | 55.22 | 78.06 | 82.36 | 77.08 | 79.17 | 65.55 | 70.45 | 61.91 | 65.97 |
| ✔ | ✘ | 77.05 | 81.62 | 70.70 | 76.46 | 76.90 | 77.42 | 76.39 | 76.90 | 80.20 | 75.15 | 75.53 | 76.96 |
| ✔ | ✔ | **79.98** | **82.17** | **83.47** | **81.87** | **95.27** | **96.82** | **96.09** | **96.06** | **92.28** | **91.74** | **89.53** | **91.18** |

**MDE Aggregation Ablation.** The Lastde consists of 2 components: Log-Likelihood and Agg-MDE. To demonstrate the indispensable role of the aggregated MDE sequence in Lastde, we conducted ablation experiments under a more complex black-box setting. As shown in Figure 10, relying solely on Log-Likelihood or Agg-MDE can only yield very limited distinctions between human and LLM texts, while their combination significantly enhances the discriminative power between the two text types. Table 11 indicates that, with appropriate aggregation functions and hyperparameter settings, relying solely on Agg-MDE can yield good results. Furthermore, when combined with Likelihood, accuracy (AUROC) can sometimes be improved by over 10%. We believe that Likelihood, as a classical statistical measure, provides a comprehensive overview of the global information in the token probability sequence. At the same time, Agg-MDE focuses on modeling segments of the token probability sequence, extracting useful yet often overlooked local information. These two types of information complement each other and are both indispensable components of Lastde.

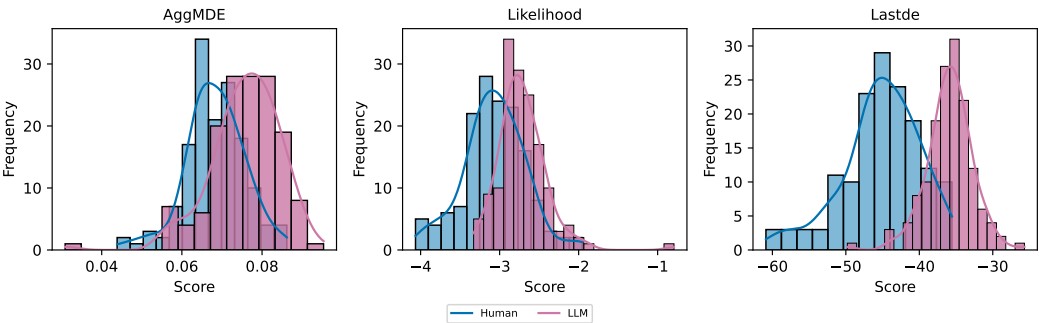

Figure 10: Black-box experiments on the WritingPrompts (GPT-2) dataset. The aggregation function and the hyperparameters for MDE both use the default settings provided in Subsection 3.2. The complete results can be found in Table 11.

**Discriminative ability of Lastde and Lastde++.** We conducted further studies on Lastde and Lastde++ across various source models. As shown in Figure 11 , the approximate threshold for distinguishing human and LLM texts using the Lastde score is -40. The distribution of human texts is relatively flat, leading to some overlap with the LLM distribution. However, Figure 12 indicates that after applying sampling normalization, the Lastde score makes it easier to distinguish between the two types of texts. The average score for LLM texts is around 4, while for human texts, it is around 0, and both distributions are more compact, resulting in only minimal overlap. In conclusion, both detection methods we propose are highly effective.

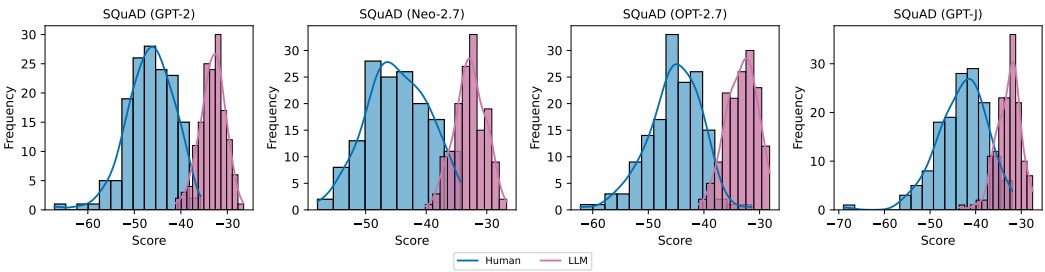

Figure 11: The distribution of white-box detection scores by *Lastde* for SQuAD, with the source models indicated in parentheses. Each subplot contains 150 pairs of LLM-generated and human-written texts. The LLM-generated texts are generated by inputting the first 30 tokens of the human-written texts as prompts into the corresponding source models B.2. The implementation settings are described in Subsection 3.2.

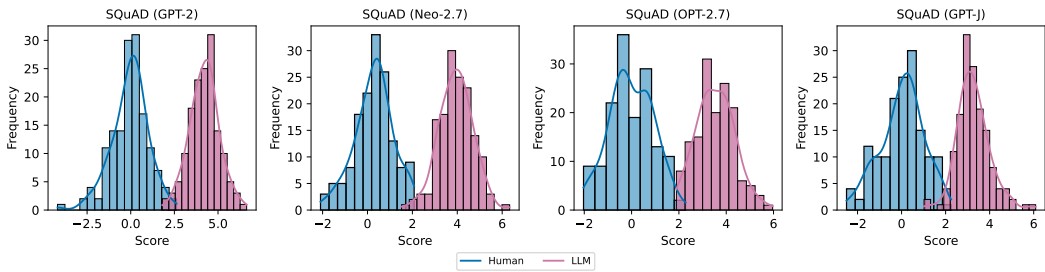

Figure 12: White-box detection score distribution of *Lastde++*. The dataset is exactly the same as that given in Figure 11.

## F  DETAILED RESULTS OF ROBUSTNESS ANALYSIS

### F.1  SAMPLES NUMBER

Previous training-free detectors typically employ three strategies for generating contrast samples: using T5 for perturbation, rewriting with the proxy or source model, and fast-sampling with the proxy or source model. This paper adopts the fast-sampling technique by default. Since Lastde can serve as a standard scoring statistic, theoretically, we can enhance Lastde using any of the three aforementioned methods. As shown in Table 12, generally speaking, the more contrast samples there are, the more accurate the scoring becomes. It is worth noting that we have already achieved state-of-the-art performance with the support of fast-sampling technique, so we did not explore the other two methods in detail. However, this does not impose any limitations on the Lastde score.

Table 12: Detailed results corresponding to Figure 4. White-box detection was performed on *XSum*, *SQuAD*, and *WritingPrompts*, with the source model being *Gemma-7* and the number of contrast samples set to $\{10, 20, 50, 100\}$.

| Methods | Datasets → | XSum | | | | SQuAD | | | | WritingPrompts | | | |
|---|---|---|---|---|---|---|---|---|---|---|---|---|---|
| ↓ | Numbers→ | 10 | 20 | 50 | 100 | 10 | 20 | 50 | 100 | 10 | 20 | 50 | 100 |
| DetectGPT | | 61.40 | 65.36 | 69.45 | 71.64 | 68.60 | 70.90 | 73.86 | 76.71 | 62.77 | 66.49 | 72.64 | 72.44 |
| DetectNPR | | 76.83 | 77.10 | 78.36 | 77.83 | 78.41 | 79.39 | 80.32 | 79.78 | 75.49 | 77.97 | 77.17 | 77.23 |
| DNA-GPT | | 58.18 | 58.43 | 59.24 | 59.78 | 67.01 | 65.69 | 67.17 | 67.68 | 62.92 | 62.53 | 63.72 | 63.07 |
| Fast-DetectGPT | | 95.51 | 95.85 | 96.27 | 96.28 | 95.67 | 96.28 | 96.82 | 96.72 | 96.74 | 96.95 | 97.27 | 97.32 |
| **Lastde++** | | **97.18** | **97.61** | **97.92** | **97.98** | **98.15** | **98.40** | **98.88** | **98.80** | **97.69** | **97.83** | **98.26** | **98.42** |

### F.2  PARAPHRASING ATTACK

Evading detection through paraphrasing attacks is a current research challenge. As shown in Table 13 and Table 14, the performance of Fast-DetectGPT drops significantly in both scenarios, while Lastde++ remains relatively stable. Overall, the white-box scenario better highlights the robustness of Lastde++. In the black-box scenario, Lastde++ performs slightly worse than Fast-DetectGPT on the

Table 13: Comparison of detection effects before and after paraphrasing attack in white-box scenario. The source models are *Llama1-13*, *OPT-13*, and *GPT-J*. The average of the three is shown in Table 3. The values in brackets indicate the decrease in AUROC after attack compared to before attack.

| Methods/Datasets | XSum(Original) | XSum(Paraphrased) | WritingPrompts(Original) | WritingPrompts(Paraphrased) | Reddit(Original) | Reddit(Paraphrased) |
|---|---|---|---|---|---|---|
| *Llama-13* | | | | | | |
| Fast-DetectGPT | 94.80 | 81.12 (↓ 13.68) | 98.35 | 95.01 (↓ 3.34) | 96.84 | 91.22 (↓ 5.62) |
| **Lastde++** | **97.27** | **85.65 (↓ 11.62)** | **99.56** | **97.69 (↓ 1.87)** | **98.48** | **95.15 (↓ 3.33)** |
| *OPT-13* | | | | | | |
| Fast-DetectGPT | 95.19 | 85.43 (↓ 9.76) | 99.40 | **97.46 (↓ 1.94)** | 98.81 | 96.61 (↓ 2.20) |
| **Lastde++** | **96.94** | **88.30 (↓ 8.64)** | 99.40 | 97.43 (↓ 1.97) | **99.43** | **98.00 (↓ 1.43)** |
| *GPT-J* | | | | | | |
| Fast-DetectGPT | 98.12 | 89.88 (↓ 8.24) | 99.22 | 96.40 (↓ 2.82) | 98.58 | 95.97 (↓ 2.61) |
| **Lastde++** | **99.04** | **93.87 (↓ 5.17)** | **99.75** | **98.41 (↓ 1.34)** | **99.38** | **97.85 (↓ 1.53)** |

Table 14: Comparison of detection effects before and after paraphrasing attack in black-box scenario. The source models are *Llama1-13*, *OPT-13*, *GPT-4-Turbo*, and the rest of the information is the same as in the Table 13

| Methods/Datasets | XSum(Original) | XSum(Paraphrased) | WritingPrompts(Original) | WritingPrompts(Paraphrased) | Reddit(Original) | Reddit(Paraphrased) |
|---|---|---|---|---|---|---|
| *Llama-13* | | | | | | |
| Fast-DetectGPT | 64.79 | 53.91 ($\downarrow$ **10.88**) | 88.26 | 82.09 ($\downarrow$ 6.17) | 79.68 | 74.05 ($\downarrow$ **5.63**) |
| Lastde++ | **72.84** | **61.27** ($\downarrow$ 11.57) | **94.46** | **90.05** ($\downarrow$ **4.41**) | **87.70** | **81.68** ($\downarrow$ 6.02) |
| *OPT-13* | | | | | | |
| Fast-DetectGPT | 84.27 | 72.63 ($\downarrow$ 11.64) | 89.64 | 88.44 ($\downarrow$ 1.20) | 84.57 | 83.73 ($\downarrow$ **0.84**) |
| Lastde++ | **91.76** | **80.58** ($\downarrow$ **11.18**) | **95.25** | **94.24** ($\downarrow$ **1.01**) | **93.09** | **91.27** ($\downarrow$ 1.82) |
| *GPT-4-Turbo* | | | | | | |
| Fast-DetectGPT | 80.79 | 74.96 ($\downarrow$ 5.83) | **89.88** | 80.93 ($\downarrow$ 8.95) | **93.87** | **91.35** ($\downarrow$ 2.52) |
| Lastde++ | **83.12** | **78.44** ($\downarrow$ **4.68**) | 88.50 | **81.32** ($\downarrow$ **7.18**) | 93.00 | 90.74 ($\downarrow$ **2.26**) |

original data, but it surpasses Fast-DetectGPT on paraphrased texts with a smaller performance drop. A similar trend can be observed on the Reddit dataset. In conclusion, Lastde++ holds significant potential for further research.

## F.3 DECODING STRATEGY

Table 15: Specific details of the Figure 6.

| Methods Strategies → | Top-$p$ ($p = 0.96$) | | | | Top-$k$ ($k = 40$) | | | | Temperature ($T = 0.8$) | | | |
|---|---|---|---|---|---|---|---|---|---|---|---|---|
| ↓ Models → | GPT-2 | Neo-2.7 | OPT-2.7 | Avg. | GPT-2 | Neo-2.7 | OPT-2.7 | Avg. | GPT-2 | Neo-2.7 | OPT-2.7 | Avg. |
| *XSum* | | | | | | | | | | | | |
| Likelihood | 68.60 | 64.04 | 81.53 | 71.39 | 59.55 | 54.47 | 73.61 | 87.67 | 62.54 | 87.07 | 91.66 | 88.80 |
| LogRank | 71.44 | 68.75 | 83.03 | 74.41 | 64.52 | 61.12 | 77.04 | 67.56 | 90.29 | 89.95 | 93.51 | 91.25 |
| DetectLRR | 73.20 | 74.52 | 80.48 | 76.07 | 73.28 | 73.65 | 79.10 | 75.34 | 91.17 | 90.28 | 93.49 | 91.65 |
| **Lastde** | **90.23** | **91.71** | **92.35** | **91.43** | **89.65** | **84.59** | **88.92** | **87.72** | **96.48** | **96.70** | **96.28** | **96.49** |
| Fast-DetectGPT | 93.57 | 90.00 | 94.07 | 92.55 | 85.19 | 84.59 | 87.93 | 85.90 | 99.28 | 98.89 | 99.31 | 99.16 |
| **Lastde++** | **97.36** | **95.62** | **97.81** | **96.93** | **93.99** | **92.95** | **95.66** | **94.20** | **99.51** | 98.88 | 99.20 | **99.20** |
| *SQuAD* | | | | | | | | | | | | |
| Likelihood | 58.24 | 63.84 | 68.61 | 63.56 | 48.60 | 54.70 | 60.24 | 54.51 | 80.71 | 85.60 | 87.00 | 84.44 |
| LogRank | 64.02 | 69.40 | 73.55 | 68.99 | 56.76 | 62.11 | 67.32 | 62.06 | 85.72 | 90.08 | 90.85 | 88.88 |
| DetectLRR | 77.24 | 80.86 | 81.96 | 80.02 | 77.78 | 80.73 | 81.80 | 80.10 | 92.16 | 94.81 | 94.49 | 93.82 |
| **Lastde** | **88.27** | **91.32** | **92.57** | **90.72** | **86.85** | **88.22** | **88.47** | **87.85** | **95.14** | **95.13** | **96.19** | **95.49** |
| Fast-DetectGPT | 93.53 | 95.96 | 94.30 | 94.60 | 88.26 | 90.22 | 90.07 | 89.52 | 98.79 | 99.73 | 99.42 | 99.31 |
| **Lastde++** | **96.84** | **98.78** | **96.98** | **97.53** | **95.08** | **96.47** | **96.48** | **96.01** | **99.02** | 99.44 | **99.65** | **99.37** |
| *WritingPrompts* | | | | | | | | | | | | |
| Likelihood | 87.87 | 90.21 | 85.00 | 87.69 | 82.16 | 85.82 | 79.08 | 82.35 | 96.38 | 97.41 | 94.59 | 96.13 |
| LogRank | 90.00 | 92.31 | 88.02 | 90.11 | 86.12 | 89.02 | 84.50 | 86.55 | 97.50 | 98.20 | 96.53 | 97.41 |
| DetectLRR | 91.93 | 95.16 | 91.28 | 92.79 | 91.65 | 93.30 | 91.40 | 92.12 | 98.08 | 98.45 | 97.14 | 97.89 |
| **Lastde** | **97.83** | **99.30** | **97.58** | **98.24** | **96.18** | **98.35** | **96.59** | **97.04** | **99.12** | **99.84** | **99.30** | **99.42** |
| Fast-DetectGPT | 98.78 | 98.83 | 96.89 | 98.17 | 96.72 | 95.01 | 92.46 | 94.73 | 99.68 | **99.80** | 98.97 | 99.48 |
| **Lastde++** | **99.68** | **99.70** | **98.86** | **99.41** | **98.84** | **98.44** | **98.00** | **98.43** | **99.76** | 99.76 | **99.33** | **99.62** |

As shown in the Table 15, Lastde++ consistently maintains a detection performance above 94% across all three datasets and outperforms Fast-DetectGPT in every strategy. Additionally, Lastde demonstrates the best average detection performance among similar methods. Notably, with a top-$k$ value of 40, Lastde++ achieves average absolute advantages over Fast-DetectGPT of 8.30%, 6.49%, and 3.70% across the three datasets, highlighting its superior robustness in handling different decoding strategies.

## F.4 DETECTING LLM-GENERATED TEXT THAT MIMICS HUMAN WRITING STYLE

In this part, we also compared the performance of Lastde and Lastde++ with other baselines in detecting LLM-generated text that mimics human writing styles. Specifically, we constructed two new datasets using *WritingPrompts* with `gpt-4-turbo-04-09` and `gpt4o-08-06`, respectively. Unlike previous work, we designed the prompts as follows: **"Please mimic the human writer an article with about 200 words starting exactly with: *<prefix>*,"** explicitly incorporating the requirement to imitate human style. The table below shows the results based on the average of three metrics (AUROC,TPR,Accuracy) across the two datasets:

Table 16: The average of *WritingPrompts* (`gpt-4-turbo-0409`, `gpt4o-08-06`).

|  | Likelihood | LogRank | DetectLRR | Lastde | Fast-DetectGPT | **Lastde++** |
|---|---|---|---|---|---|---|
| AUROC | 87.37 | 85.60 | 74.94 | 87.68 | **96.49** | **96.49** |
| TPR at 5% FPR | 30.00 | 30.67 | 24.34 | 29.33 | 82.67 | **85.36** |
| Accuracy at 5% FPR | 62.50 | 62.83 | 59.67 | 63.34 | 88.84 | **90.17** |

For both human-style mimicking datasets, Lastde++ shows comparable or better results across all three metrics compared to Fast-DetectGPT  (Bao et al., 2024), demonstrating its effectiveness in detecting LLM text that mimics human writing style.

## F.5 DETECTING LLM-LLM MIXTURE TEXT

In real-world scenarios, malicious users may generate text using multiple different LLMs. Therefore, it is necessary to explore the performance of existing detectors on a type of text referred to as LLM-LLM mixture text. To this end, we constructed a dataset of texts generated by multiple different LLMs, using **WritingPrompts** as an example. Specifically, we conducted detection experiments on text mixtures generated by two and four LLMs.

In the two-model mixture experiments, we evaluated two different mixing ratios: 50%+50% (*Llama2-13B*, *OPT-13B*) and 80%+20% (*BLOOM-7B*, *Falcon-7B*). In the four-model mixture experiments, we used texts with an equal distribution of 25% from each of four LLMs (*Llama2-13B*, *OPT-13B*, *BLOOM-7B*, *Falcon-7B*).

We employed a simple "truncation-concatenation" method to generate the mixed data. For instance, we took 50% of the characters from the text generated by the first source model and concatenated it with 50% of the text from the second model. Although this approach might lose some contextual information, we believe it is sufficient and effective for demonstration purposes. The experimental results are as follows:

Table 17: AUROC on LLM-LLM mixture text

| Mixing ratio | Source Models | Likelihood | LogRank | DetectLRR | Lastde | Fast-DetectGPT | Lastde++ |
|---|---|---|---|---|---|---|---|
| 50%+50% | (Llama2-13B, OPT-13B) | 81.48 | 84.81 | 87.99 | 91.88 | 85.21 | **92.48** |
|  | (BLOOM-7B, Falcon-7B) | 78.80 | 82.56 | 87.47 | 94.95 | 88.59 | **95.65** |
| 80%+20% | (Llama2-13B, OPT-13B) | 80.44 | 82.97 | 85.45 | 89.13 | 86.73 | **92.64** |
|  | (BLOOM-7B, Falcon-7B) | 76.98 | 81.52 | 52.91 | 96.38 | 92.05 | **96.57** |
| 25% each | (Llama2-13B, OPT-13B, BLOOM-7B, Falcon-7B) | 82.33 | 86.34 | 88.02 | **94.44** | 86.02 | 93.32 |

Table 18: TPR at 5% FPR on LLM-LLM mixture text

| Mixing ratio | Source Models | Likelihood | LogRank | DetectLRR | Lastde | Fast-DetectGPT | Lastde++ |
|---|---|---|---|---|---|---|---|
| 50%+50% | (Llama2-13B, OPT-13B) | 29.33 | 36.00 | 40.00 | 65.33 | 46.67 | **68.00** |
|  | (BLOOM-7B, Falcon-7B) | 16.00 | 30.00 | 34.67 | **76.00** | 39.33 | 70.67 |
| 80%+20% | (Llama2-13B, OPT-13B) | 24.67 | 28.67 | 38.67 | 58.67 | 41.33 | **69.33** |
|  | (BLOOM-7B, Falcon-7B) | 16.67 | 27.67 | 35.33 | 59.67 | 49.33 | **83.33** |
| 25% each | (Llama2-13B, OPT-13B, BLOOM-7B, Falcon-7B) | 22.00 | 40.67 | 46.67 | **81.33** | 40.67 | 73.33 |

Table 19: Accuracy at 5% FPR on LLM-LLM mixture text

| Mixing ratio | Source Models | Likelihood | LogRank | DetectLRR | Lastde | Fast-DetectGPT | Lastde++ |
|---|---|---|---|---|---|---|---|
| 50%+50% | (Llama2-13B, OPT-13B) | 62.17 | 65.50 | 67.50 | 80.17 | 70.83 | **81.50** |
|  | (BLOOM-7B, Falcon-7B) | 55.55 | 62.50 | 64.83 | 85.50 | 67.17 | **82.83** |
| 80%+20% | (Llama2-13B, OPT-13B) | 59.83 | 61.83 | 66.83 | 76.83 | 68.17 | **82.17** |
|  | (BLOOM-7B, Falcon-7B) | 55.83 | 58.83 | 69.83 | 87.83 | 77.17 | **89.17** |
| 25% each | (Llama2-13B, OPT-13B, BLOOM-7B, Falcon-7B) | 58.50 | 67.83 | 70.50 | **88.17** | 67.83 | 84.17 |

The results show that even with texts mixed from different source models and different ratios, **Lastde** and **Lastde++** still achieve good detection performance. In all three metrics across all datasets, Lastde or Lastde++ reach SOTA, especially with TPR (at 5% FPR) exceeding Fast-DetectGPT, the prior SOTA, by at least 20%.

## G  DISCUSSION ON THRESHOLD SELECTION IN REAL-WORLD SCENARIOS

For training-free detectors, most studies  Su et al. (2023); Bao et al. (2024); Yang et al. (2024) use AUROC as the evaluation metric, which does not require setting a specific threshold. However, real-world applications often require decision-making at a fixed threshold, so it is essential to examine the performance of detectors under specific thresholds. We investigated two methods:

**The perspective of Fast-DetectGPT.** The authors  (Bao et al., 2024) of the paper have released a local inference code for making decisions on individual samples. In this code, they use an "*offset*" to estimate the threshold on a specific dataset (e.g., XSum(ChatGPT)), and then apply it to other datasets. The authors claim that this estimation method does not introduce significant bias.

**Perspectives on some benchmarks.** Other related studies  (He et al., 2024; Mao et al., 2024; Zhang et al., 2024) achieve final classification by using detection scores and labels to fit a univariate logistic regression model.

We prefer the latter approach and have fitted logistic regression models on datasets (including Xsum, WritingPrompts, Reddit) generated by two closed-source models (GPT-4-Turbo, GPT-4o) and one open-source model (OPT-13B), reporting metrics on the test set (test size=0.2). Average AUROC/TPR metrics at 1%FPR on three datasets are reported below.

Table 20: Comparison of AUROC and TPR values on three datasets

| Metrics | Methods | GPT-4-Turbo | GPT-4o | OPT-13 | Avg. |
|---------|---------|-------------|--------|--------|------|
| AUROC | Fast-DetectGPT | 88.91 | 90.81 | 78.27 | 86.00 |
|       | Lastde++ | 87.02 | 91.29 | 95.88 | **91.40** |
| TPR at 1%FPR | Fast-DetectGPT | 26.88 | 46.24 | 24.73 | 32.62 |
|              | Lastde++ | 41.94 | 48.39 | 66.67 | **52.33** |

The fixed threshold here corresponds to the decision boundary of the logistic regression model. For closed-source models, Lastde++ shows comparable or better performance in both AUROC and TPR compared to Fast-DetectGPT, while significantly outperforming Fast-DetectGPT on open-source models. However, this threshold determination method still relies on true labels, which we consider a limitation, as it appears to undermine the 'training-free' aspect (i.e., the inability to use real labels).

To resolve this issue, we propose a more reasonable modification. According to Figure 12 in our Appendix E, regardless of the source model, the Lastde++ score boundary between human-written and LLM-generated texts consistently appears around 2.0, which can be estimated more accurately with larger sample sizes. While setting the Lastde++ detection threshold directly to 2.0 might yield satisfactory results, a more robust approach would be to first define a neighborhood with radius centered at 2.0, then assign *pseudo-labels* to samples on either side (human for left, LLM for right), and finally fit a logistic regression model. Notably, this modified approach doesn't utilize any true labels, reducing the disruption to the assumptions of the training-free detector.

