# OpenReview forum: "Training-free LLM-generated Text Detection by Mining Token Probability Sequences"
_ICLR.cc/2025/Conference — ICLR 2025 Poster_

### Official Review · Reviewer_LuhL · 2024-10-30

**Soundness:** 3
**Presentation:** 3
**Contribution:** 3
**Rating:** 6
**Confidence:** 3

**Summary:**

This paper presents an AIGC detection method named Lastde. It is training free and can used both local and global statistics for
AIGC detection. An more efficient version named Lastde++ is also proposed for real-time detection.

**Strengths:**

1. This paper presents an effective AIGC detection method, which can be very useful in the GenAI era.

2. The proposed method in sound and does not rely on training.

3. The experiments show promising results.

**Weaknesses:**

1. It would be better if the paper writing can be improved. For example, figure 2 is not so easy to understand.

2. My biggest concern is that strong baselines are not included in experiments, such as GPTZero.

3. Can the proposed method work well in different domains?

4. Is the proposed method robust to some adaptive attacks like asking LLMs to mimic the human writing?

**Questions:**

1. It would be better if the paper writing can be improved. For example, figure 2 is not so easy to understand.

2. My biggest concern is that strong baselines are not included in experiments, such as GPTZero.

3. Can the proposed method work well in different domains?

4. Is the proposed method robust to some adaptive attacks like asking LLMs to mimic the human writing?

---

> ### Author Response · Authors · 2024-11-23
> **Response to Weakness (W) / Questions (Q)**
>
> We appreciate the Reviewer LuhL for your invaluable suggestions to improve the quality of our work. We have addressed all concerns as below.
>
> **W1:** We apologize for any confusion caused by our writing. Figure 2 illustrates the computational workflow or framework of Lastde, using a 7-token text as an example. While the global statistics calculation is straightforward, for local statistics, the sequence of token probabilities (TPS) with 7 tokens is first processed into 3 new sequences (as we set $\tau^{\prime}=3$). Here, $\tau_i\in{1,2,3}$ represents the scale factor, indicating that the current scale is derived from the arithmetic mean of $\tau_i$ adjacent elements in the original TPS. For each new sequence, we slide from left to right with a stride of 1 and window width of 3 (as we set $s=3$), vertically concatenating each sliding window segment into a matrix. We then calculate the cosine similarity between adjacent rows (segments) to obtain a similarity sequence. Finally, we divide [-1,1] into 10 subintervals (as we set $\varepsilon=10$) and compute the frequency or probability of elements from the similarity sequence in each subinterval (resulting in a $1\times 10$ vector), followed by calculating the sequence entropy to obtain the DE value. The final step in local statistics calculation involves using an aggregation operator $\text{Agg}:\mathbb{R}^{\tau^{\prime}}\rightarrow\mathbb{R}$ to summarize all DE values. The corresponding mathematical derivations can be found in Section 3.1, and we have updated the main text accordingly with changes marked in red.
>
> **W2:** We appreciate this constructive comment. Previous studies such as DNA-GPT[1] (ICLR2024) and Fast-DetectGPT[2] (ICLR2024) indeed included GPTZero as a baseline. However, since our main focus was on training-free methods, we did not test any training-based detectors in the main text and instead concentrated on Fast-DetectGPT as our main comparison target. Nevertheless, we have addressed this concern by including results from **two powerful baselines** in Table R1 (in General Response): Binoculars[3] (ICML2024) and GPTZero. Here is a brief summary of the results (please refer to Table R1 for complete details).
>
> |   | Binoculars | GPTZero | **Lastde** | **Lastde++** |
> | --- | --- | --- | --- | --- |
> | without TOCSIN  | 87.5 |88.6 | 89.6 | **94.2** |
> | with TOCSIN | 87.5 | 88.6 | 93.3 | **95.3** |
> | improvement | - | - | 3.7 | 1.1 |
>
> **W3:** Indeed, the ability to achieve universal detection across different domains or subjects is a notable advantage of all training-free methods, including the proposed method. To support this claim, we have provided extensive experimental results in the paper and Table R1 (in General Response). Specifically, most experiments in the main text were conducted on four datasets: Xsum, SQuAD, WritingPrompts, and Reddit, covering news, Wikipedia context knowledge, story creation, and community Q&A scenarios. Notably, the Reddit dataset encompasses six subjects: biology, physics, chemistry, economics, law, and technique. Furthermore, we included German and Chinese datasets, with the Chinese dataset explicitly containing samples from three subjects: food, history, and economics. Table 1 also presents detection results on PubMedQA, an expert-level Q&A dataset focusing on biochemical knowledge. The results consistently demonstrate our method's superiority over other detectors, providing compelling evidences for our method's robustness to cross-domain detection.
>
> **W4:** Thank you for the great suggestion. In the rebuttal, we have conducted extensive experiments to validate the robustness of our method, and summarized the results in Table R2 (in General Response). In brief,  even when we explicitly instructed the LLM to mimic human writing styles, Lastde++ still achieved excellent detection performance. Moreover, Lastde++ demonstrates greater advantages over Fast-DetectGPT in terms of TPR and Accuracy, which are more practical metrics. Below are the TPR at 5% FPR results from Tables R2 (in General Response):
> |                                    | Likelihood | LogRank | DetectLRR | **Lastde** | Fast-DetectGPT | **Lastde++** |
> | ---------------------------------- | ---------- | ------- | --------- | ---------- | -------------- | ------------ |
> | WritingPrompts（gpt-4-turbo-0409） | 20.00      | 22.67   | 20.67     | 21.33      | 74.00          | **78.67**    |
> | WritingPrompts（gpt4o-0806）       | 40.00      | 38.67   | 28.00     | 37.33      | 91.33          | **92.00**    |
>
> [1] DNA-GPT: Divergent N-Gram Analysis for Training-Free Detection of GPT-Generated Text
>
> [2] Fast-DetectGPT: Efficient Zero-Shot Detection of Machine-Generated Text via Conditional Probability Curvature
>
> [3] Binoculars: Spotting LLMs With Binoculars: Zero-Shot Detection of Machine-Generated Text

---

> > ### Comment · Reviewer_LuhL · 2024-11-26
> >
> > Thank the authors for addressing my concerns. I hope the information in the rebuttal can be added to the final version paper.

---

> > > ### Author Response · Authors · 2024-11-26
> > > **Thanks and paper revised accordingly**
> > >
> > > Dear Reviewer LuhL,
> > >
> > > We sincerely thank you again for your constructive feedback! We are pleased to know that our rebuttal successfully addressed your initial concerns. Based on your suggestions, we have carefully revised our paper, and we believe these revisions have significantly improved the quality and presentation of our work.
> > >
> > > Below is a summary of the revisions made:
> > >
> > > - **Table 9 (Appendix D, Page 20-21):** We have added comparison results with GPTZero and other advanced baseline methods (e.g., Binoculars). These additions are highlighted in red for clarity.
> > >
> > > - **Table 16 (Appendix F4, Page 25-26):** We added the results of LLM-generated text detection that mimics human writing style to Appendix F4 and highlighted them in red.
> > >
> > > - **Figure 2 (Page 3):**  We revised the description of the Lastde frame, which is also highlighted in red.
> > >
> > >
> > > We hope these updates address any remaining questions or concerns. We remain fully committed to further improving the manuscript and welcome any additional insights or suggestions you might have. Please let us know if further clarifications or additional details are required.
> > >
> > > Warmest regards,
> > >
> > > The Authors

---

### Official Review · Reviewer_Kqgm · 2024-11-03

**Soundness:** 3
**Presentation:** 3
**Contribution:** 3
**Rating:** 6
**Confidence:** 4

**Summary:**

The paper introduces a training-free method called Lastde for detecting LLM-generated text by analyzing TPS through a blend of local and global statistical features. Lastde incorporates temporal dynamics in TPS with diversity entropy to capture subtle distinctions between human and AI-generated text. An enhanced version, Lastde++, offers faster, real-time detection and outperforms existing methods across various scenarios. Extensive experiments demonstrate that Lastde++ provides superior robustness against paraphrasing attacks and cross-lingual challenges, establishing it as a powerful benchmark in LLM-generated text detection​.

**Strengths:**

- The paper attempts to solve an important LLM-generated text detection problem without further training

- The proposed method is straightforward and intuitive, and the experiments presented in the paper are comprehensive with solid results.

- The readers can easily understand the proposed method and follow the content of the paper.

**Weaknesses:**

- I recommend including more evaluations on separate sets of LLM-generated text and human-written text ( i.e., evaluating the method on sets containing only LLM-generated text and only human-written text). This would provide valuable insights into how the proposed method works and whether the detection method performs better on LLM-generated text, or human-written text, or both.

- Paraphrasing attacks pose a significant threat in the context of LLM-generated text detection. It is highly recommended to use a more powerful LLM paraphraser, which could better highlight the proposed method’s ability to protect against such attacks.

- How did you predetermine the detection threshold, especially when the method is training-free and there is no prior knowledge about the LLM-generated text for the detector? The paper does not discuss the details of how the threshold was chosen but only provides a specific value used in a certain scenario. The threshold is a key component that leads to good detection performance no matter how good the proposed scores are extracted.

- Generally speaking, when we attempt to detect whether a given text is generated by LLMs, we usually do not know which specific models malicious users have employed to produce the text. It would be interesting to see a further experiment conducted on a mixed set of LLM-generated text created from various sources LLMs. The evaluation of detection abilities in the above experimental settings more closely matches real-world scenarios.

**Questions:**

Please refer to the weakness.

---

> ### Author Response · Authors · 2024-11-23
> **Response to Weakness (W) : Part (1/2)**
>
> We appreciate the reviewer Kqgm for your constructive suggestions, we will address all raised concerns point by point as below.
>
> **W1:** Thank you for the insights. In the revision, we additionally report the TPR and TNR values to address this concern. More specifically, we consider human-written text as the negative class and LLM-generated text as the positive class. When fixing FPR = 5% (i.e., TNR = 95%), we observe the following TPR results:
>
> | **Datasets/Methods**       | **Likelihood** | **LogRank** | **DetectLRR** | **Lastde** | **Fast-DetectGPT** | **Lastde++** |
> | -------------------------- | -------------- | ----------- | ------------- | ---------- | ------------------ | ------------ |
> | WritingPrompts(Llama2-13b) | 50.17          | 50.50       | 59.17         |**64.50**      | 54.83              | **74.50**        |
> | WritingPrompts(OPT-13b)    | 52.17          | 54.17       | 62.17         | **76.83**  | 59.50              | **76.17**    |
>
> These results demonstrate that Lastde++ achieves TPR values higher than 74% on both benchmark datasets, indicating that while correctly classifying 95% of human-written texts, it can also accurately identify at least 74% of LLM-generated texts. This performance significantly outperforms other training-free methods. Therefore, we believe Lastde++ demonstrates robust capability in discriminating both human-written and LLM-generated texts.
>
> **W2:** We have selected two representative datasets and conducted additional tests using a paraphraser with larger parameter count. The details about this paraphraser and comprehensive detection results are reported in Table R3 (in General Response). In brief, even when texts are attacked using more powerful models, our method still achieves SOTA performance. Furthermore, Tables R2 - R6 contain additional robustness test results, which we believe will help address your concerns.
>
> **W3:** We apologize for not elaborating on the threshold determination method in our paper. For training-free detectors, most studies use AUROC as the evaluation metric, which doesn't require fixing a specific threshold. However, as the reviewer pointed out, practical applications typically require decision-making at a specific threshold. We have examined two approaches:
> - Fast-DetectGPT [1] (ICLR2024) uses an "offset" to estimate thresholds on specific datasets (e.g., XSum(GPT-4)) in their open-source code (https://github.com/baoguangsheng/fast-detect-gpt/blob/main/scripts/local_infer.py).
>
> - Other related studies, such as MGTBench [2] (CORR2023) and RAIDAR [3] (ICLR2024), fit a univariate logistic regression model using both detection scores and labels for final classification.
>
> We prefer the latter approach and have fitted logistic regression models on datasets (including Xsum, WritingPrompts, Reddit) generated by two closed-source models (GPT-4-Turbo, GPT-4o) and one open-source model (OPT-13B), reporting metrics on the test set (test size=0.2). Average AUROC/TPR metrics at 1%FPR on three datasets are reported below.
>
> | Metrics | Methods | GPT-4-Turbo | GPT-4o | OPT-13 | Avg. |
> | --- | --- | --- | --- | --- | --- |
> | AUROC | Fast-DetectGPT | 88.91 | 90.81 | 78.27 | 86.00 |
> |  | Lastde++ | 87.02 | 91.29 | 95.88 | **91.40** |
> | TPR at 1%FPR | Fast-DetectGPT | 26.88 | 46.24 | 24.73 | 32.62 |
> |  | Lastde++ | 41.94 | 48.39 | 66.67 | **52.33** |
>
> The fixed threshold here corresponds to the decision boundary of the logistic regression model. For closed-source models, Lastde++ shows comparable or better performance in both AUROC and TPR compared to Fast-DetectGPT, while significantly outperforming Fast-DetectGPT on open-source models. However, this threshold determination method still requires true labels, which we acknowledge as a limitation (not entirely training-free).
>
>  To resolve this issue, we propose a **more reasonable modification**: According to Figure 12 in our Appendix E, regardless of the source model, the Lastde++ score boundary between human-written and LLM-generated texts consistently appears around 2.0, which can be estimated more accurately with larger sample sizes. While setting the Lastde++ detection threshold directly to 2.0 might yield satisfactory results, a more robust approach would be to first define a neighborhood with radius $r$ centered at 2.0, then assign pseudo-labels to samples on either side (human for left, LLM for right), and finally fit a logistic regression model. Notably, this modified approach doesn't utilize any true labels, maintaining its training-free nature.

---

> ### Author Response · Authors · 2024-11-23
> **Response to Weakness (W) : Part (2/2)**
>
> **W4:** We agree with the reviewer's insight and have constructed new datasets as suggested. These datasets combine texts generated by multiple source models, incorporating various mixing ratios and model numbers. Detailed experimental procedures, workflows, and metrics are presented in Tables R4-R6 (in General Response). In summary, both Lastde and Lastde++ demonstrate effective detection capabilities across different scenarios, whether combining 2 or 4 source models, or mixing with ratios of 50%+50% or 80%+20%. Their detection results (AUROC, TPR, Accuracy) **consistently outperform existing training-free detectors** like Fast-DetectGPT. We hope these results can help address your concerns.
>
> [1] Fast-DetectGPT: Efficient Zero-Shot Detection of Machine-Generated Text via Conditional Probability Curvature
>
> [2] MGTBench: Benchmarking Machine-Generated Text Detection
>
> [3] Raidar: geneRative AI Detection viA Rewriting

---

> > ### Comment · Reviewer_Kqgm · 2024-11-27
> > **Thanks for you responses**
> >
> > I would like to thank the authors for the detailed responses to my questions. Please be sure to include the discussion in the manuscript, especially the threshold part. I will maintain my score.

---

> > > ### Author Response · Authors · 2024-11-28
> > > **Thanks and paper revised accordingly**
> > >
> > > We are pleased to know that our rebuttal has alleviated your concerns. We have included the discussion in the manuscript, particularly about the selection of threshold.
> > >
> > > The following updates have been added in the latest revision:
> > >
> > > - **Table 9-10 (Appendix D, Page 20-21):** We have added comparisons of more baselines across additional datasets and source models, with all results and discussions highlighted in red.
> > > - **Table 16-19 (Appendix F.4, Page 25-26):** We have added more robustness results and detailed metrics, including the detection of LLM-generated texts that mimic human writing styles, detection of mixed texts from different source models, and intuitive evaluation metrics such as TPR and Accuracy (at 5% FPR). All additions are highlighted in red.
> > > - **Table 20 (Appendix G, Page 27):** We have created a dedicated section in Appendix G to clearly present the threshold determination method. This section illustrates how training-free detectors can be implemented in practical scenarios.
> > >
> > > Thank you again for your constructive feedback, especially regarding threshold selection and determination. We believe these improvements will greatly facilitate future research.
> > >
> > > Best regards,
> > >
> > > The Authors

---

### Official Review · Reviewer_WrSh · 2024-11-04

**Soundness:** 2
**Presentation:** 3
**Contribution:** 2
**Rating:** 6
**Confidence:** 4

**Summary:**

The paper presents a training-free detector for LLM-generated text by analyzing token probability sequences. The proposed Lastde method combines both global and local statistics, using diversity entropy to capture temporal dynamics, thus achieving improved cross-domain detection and robustness against paraphrasing attacks. Evaluations on 6 datasets show that Lastde outperforms existing detectors.

**Strengths:**

- The method is highly effective across cross-domain and cross-model scenarios without needing retraining.
- Lastde++ enables real-time detection with minimal computational cost, outperforming many established methods.
- Exhibits strong resistance to paraphrasing attacks, maintaining accuracy in varied textual manipulations.

**Weaknesses:**

- The detection accuracy depends significantly on the chosen proxy model, especially in black-box settings, affecting usability across model types.
- The choice of proxy model in black-box scenarios is still unclear, and the use of GPT-j need more justification.
- Performance drops with shorter texts, which limits applicability in scenarios like social media or short Q&A responses.

**Questions:**

How does Lastde handle highly stylized human-written text that might mimic AI-generated patterns?
Can Lastde be adapted or optimized for human-LLM co-authored text (e.g., LLM-revised text)?
Would a hybrid approach combining Lastde with a lightweight training-based model further enhance detection accuracy?

---

> ### Author Response · Authors · 2024-11-23
> **1) Response to Weakness (W)**
>
> We appreciate the reviewer WrSh for recognizing the effectiveness of our method and we would like to address all raised concerns in detail.
>
> - **W1:** As the reviewer pointed out, the selection of proxy models may significantly impacts the performance of black-box detection, which is a common challenge encountered by almost all training-free detectors. Recent work such as DALD [1] (NeurIPS 2024) has attempted to align proxy models with source models through distillation, partially alleviating this pain point. However, selecting more general and generalizable black-box proxy models in a principled may remains challenging. Our proposed method focuses on extracting inherent textual features and represents foundational work that can be readily integrated with recent detection methods (such as DALD). Moreover, as demonstrated with our extensive experiments, the proposed method can consistently achieve state-of-the-art performances across all datasets, including the black-box scenarios, even though the proxy model selection was not explicitly optimized. And we believe its detection effectiveness can be further improved using a more carefully selected proxy model.
>
> - **W2:** We apologize for the confusion regarding the choice of proxy model.
>    - **a):** We would like to clarify that, in Section 4.1 "Source models & proxy model" and Appendix B.3, we have detailed the proxy model selection criteria for all detection methods. Unless otherwise specified, all detection methods use GPT-J as the sole proxy model. Specifically, for "Sample-based" methods, we directly use GPT-J for scoring without additional processing. For "Distribution-based" methods, DetectGPT [2] (ICML 2023) and DetectNPR [3] (EMNLP 2023) first use T5-3B to generate perturbed samples, then use GPT-J to score the distribution formed by perturbed and original texts. For DNA-GPT [4] (ICLR 2024), we first truncate candidate texts, input the first half into GPT-J to generate rewritten samples, and finally use GPT-J to score the distribution formed by rewritten and original texts. Fast-DetectGPT [5] (ICLR 2024) and Lastde++ first use GPT-J for inference on candidate texts, then perform fast sampling on the resulting logits to obtain contrastive samples. Finally, GPT-J scores the distribution formed by contrastive and original texts.
>
>    - **b):** In Fast-DetectGPT, the prior SOTA, it explored GPT-2, Neo-2.7B, and GPT-J as proxy models, with Neo-2.7B as the default scoring model and GPT-J as the sampling/rewriting model (if applicable), and T5-3B as the perturbation model (if applicable). Following this setup, we conducted experiments on GPT-2, Neo-2.7B, and GPT-J, finding that using GPT-J for scoring, sampling, and rewriting yielded optimal results on our datasets, hence our default configuration. However, as mentioned in the previous paragraph, selecting black-box proxy models remains extremely challenging. To the best of our knowledge, few studies explicitly guide proxy model selection, necessitating reliance on detection results and generalization experiments to infer more universally applicable proxy models. In Section 4.3 "Selection of proxy models," we discussed detection results across different proxy models for various source texts. While these results can guide proxy model selection to some extent, we agree that more theoretical and empirical analysis is needed to establish selection criteria.
>
> - **W3:** As pointed out by the reviewer, detection performance decreases with shorter texts, a challenge faced by all detection methods (both training-free and training-based). While MPU [6] reformulates AI text detection as a Positive-Unlabeled (PU) learning problem and shows promising results on short-text datasets like HC3 and TweepFake, substantial room for improvement remains. This is parimarily because short texts contain limited information and often lack detectable tendencies (human or AI). Therefore, we believe that detecting short texts (e.g., in social media scenarios, AI-generated rumors, fake news) may require combining textual features (such as likelihood, rank, Lastde) with external tools. This represents both potential value in our method and a direction for our future exporation.
>
> [1] DALD: Improving Logits-based Detector without Logits from Black-box LLMs
>
> [2] DetectGPT: Zero-Shot Machine-Generated Text Detection using Probability Curvature
>
> [3] DetectLLM: Leveraging Log Rank Information for Zero-Shot Detection of Machine-Generated Text
>
> [4] DNA-GPT: Divergent N-Gram Analysis for Training-Free Detection of GPT-Generated Text
>
> [5] Fast-DetectGPT: Efficient Zero-Shot Detection of Machine-Generated Text via Conditional Probability Curvature
>
> [6] MPU: Multiscale Positive-Unlabeled Detection of AI-Generated Texts

---

> ### Author Response · Authors · 2024-11-23
> **2) Response to Questions (Q)**
>
> - **Q1:** We would like to clarify that both Fast-DetectGPT [1] and DetectGPT [2] share similar fundamental assumptions about human and LLM-generated texts: given a context, LLMs tend to use tokens with higher probabilities in the next-token prediction, while humans rely more on semantics rather than probabilities. Under this assumption, we believe it is challenging for humans to precisely mimic AI writing styles. During the rebuttal, while we have tried our best to find datasets containing human-written texts mimicking AI styles, we haven't found a suitble one (we would promptly include these experiments if the reviewer could help specify such datasets). Nevertheless, as a compensatory measure, we constructed LLM-generated texts mimicking human writing styles, with results shown in Table R2 (in General Response). These results demonstrate that Lastde++ is comparable or better than Fast-DetectGPT in AUROC, TPR, and Accuracy metrics.
>
> - **Q2:** We have addressed this concern and provided additional experiments in the **Detecting LLM/human mixted texts** in **Response to Concern II** in our General Response. In brief, we simulated human modifications using a paraphraser to attack LLM-generated texts. These attacked texts can be viewed as human-LLM collaborative generations. Detection results on these attacked texts show that Lastde++ outperforms Fast-DetectGPT with stronger robustness (smaller performance degradation). Additionally, when using the paraphraser to attack both human and LLM-generated texts, which we consider as LLM-Revised texts, Lastde++ demonstrates superior performance and robustness compared to Fast-DetectGPT.
>
> - **Q3:** Thank you for the insightful observation.
>   - **a):** We notice some training-based studies that use token log probability as initial features (such as Sniffer [3], SeqXGPT [4] (EMNLP 2023), POGER (IJCAI 2024)). However, Lastde's final output, after aggregating global and local statistics, is a real number (similar to likelihood and rank metrics). From a feature extraction perspective (e.g., "probabilistic features" in Section 2), the dimensionality or information content might seem insufficient. Since our goal is to develop a universal detector with strong generalization capabilities (a characteristic strength of training-free methods), we prefer combining Lastde with plug-and-play lightweight modules like DALD [5] (NeurIPS 2024) and TOCSIN [6] (EMNLP 2024), enhancing detection performance while maintaining the significant cross-domain, cross-model, and cross-lingual advantages of training-free methods.
>
>   - **b):** In the revision, we have combined Lastde, Lastde++, and existing baselines with the TOCSIN lightweight model, with results shown in Table R1 (in General Response). The results clearly demonstrate performance improvements across all detection methods when combined with TOCSIN. Lastde and Lastde++ showed average improvements of **3.7%** and **1.1%** respectively, achieving SOTA performance both before and after integration, which we believe strongly validates your observation. We apprecite your insights again, and we have included these results in the Appendix.
>
> [1] Fast-DetectGPT: Efficient Zero-Shot Detection of Machine-Generated Text via Conditional Probability Curvature
>
> [2] DetectGPT: Zero-Shot Machine-Generated Text Detection using Probability Curvature
>
> [3] Sniffer: Origin Tracing and Detecting of LLMs
>
> [4] SeqXGPT: Sentence-Level AI-Generated Text Detection
>
> [5] POGER: Ten Words Only Still Help: Improving Black-Box AI-Generated Text Detection via Proxy-Guided Efficient Re-Sampling
>
> [6] DALD: Improving Logits-based Detector without Logits from Black-box LLMs
>
> [7] TOCSIN: Zero-Shot Detection of LLM-Generated Text using Token Cohesiveness

---

> > ### Comment · Reviewer_WrSh · 2024-11-26
> >
> > Thank you for the detailed response to the questions. While your answer addressed some of my concerns, the limitations mentioned in the Weaknesses section still persist. Therefore, I would prefer to maintain the current score.

---

> ### Author Response · Authors · 2024-11-28
> **Thanks for feedback, please check our further clarification and revisions**
>
> We sincerely thank Reviewer WrSh for your thoughtful feedback, and we are encouraged that our rebuttal has alleviated your concerns. We have incorporated new results, analyses, and discussions into our revised manuscript, which we believe significantly enhanced the clarity and overall quality of our work.
>
> We appreciate the Reviewer’s insightful observations regarding potential challenges in scenarios such as proxy model selection in black-box settings and short-text detection. We acknowledge that these are indeed important and promising research directions, both for our future work and for advancing this field more broadly.
>
> Meanwhile, we would like to clarify that these challenges are long-standing and prevalent in the domain of AI text detection. Despite these inherent difficulties, as demonstrated in our work (and summarized in the General Response), our proposed method consistently achieves state-of-the-art performance among training-free approaches across diverse scenarios, including cross-domain, cross-model, and cross-lingual detection tasks, under both white-box and black-box settings.
>
>
> Notably, as detailed in Table R1 of the General Response, our approach can be augmented with external tools to further improve our performance, even surpassing commercial detectors like GPTZero.
>
> We hope this response addresses any remaining concerns and would greatly appreciate any additional suggestions or insights you may have.
>
> Thank you again for your constructive feedback and support.
>
> Sincerely,
>
> The Authors

---

### Official Review · Reviewer_SbPp · 2024-11-06

**Soundness:** 3
**Presentation:** 4
**Contribution:** 3
**Rating:** 6
**Confidence:** 5

**Summary:**

This paper introduces a novel training-free detector, termed Lastde that synergizes local and global statistics for enhanced detection. They introduce time series analysis to LLM-generated text detection, capturing the temporal dynamics of token probability
sequences. By integrating these local statistics with global ones, their detector reveals significant disparities between human and LLM-generated texts. They also propose an efficient alternative, Lastde++ to enable real-time detection. Extensive experiments on six datasets involving cross-domain, cross-model, and cross-lingual detection scenarios, under both white-box and black-box settings, demonstrated
that their method consistently achieves state-of-the-art performance. Furthermore, their approach exhibits greater robustness against paraphrasing attacks compared to existing baseline methods.

**Strengths:**

originality: treating text detection as time series analysis is novel and the approach seems promising, though previous work SeqXGPT already did this.

quality: I like the solid experiments from this paper since it compared with many previous baselines on a bunch of models and datasets. From all the experimental results, we can see the great potential of this method.

clarity: This paper is well-written and easy to follow. The tables and figures and clear.

significance: This provides a new method for detection and potentially useful for detection.

**Weaknesses:**

AUROC is not a very good measure for real-world use because FPR in real life is very important for this task, especially for students. I would like to know the TPR rate at 1% FPR.

In my opinion, treating the detection as a time series analysis is not entirely new. For example, your cited another work SeqXGPT also treats the token logits as waves, though your approach is different. Also, there is no comparison with SeqXGPT.

There are many hyperparameters in this algorithm to tune, weakening its potential practical usage.

Many tables rely on results on very small models like GPT-2 Neo-2.7 OPT-2.7, which makes it less significant.

**Questions:**

1. I have one question. I understand that this is a training-free method. But it seems it includes various ingenious designs such as multiscale log-probability sequence, sliding-segmentation and diversity entropy and etc. And I would assume this can not be designed at the first glance. So thinking from the first principle, given the token logits, if the token logtis can be a good indicator through another function f(), then what's the best f()? perhaps we can directly a network to find the f()?

2. It is unclear to me whether the used models like llama are the base model or chat model because instruction-tuned models are more difficult to detect and base models are easy.

3. Do you need to know the prompt and question used for generating the text for detection? in reality, we do not know this.

---

> ### Author Response · Authors · 2024-11-23
> **1) Response to Weakness (W) : Part (1/2)**
>
> We thank Reviewer SbPp for your thorough review of our paper. We would like to address your concerns, raised in Weakness and Questions parts, as below.
>
> - **W1:** We agree with your point. While AUROC may be be not a comprehensive metric, we followed the conventional practice from popular baseline methods, including Fast-DetectGPT[1] (ICLR2024), DetectGPT[2] (ICML2023), and DetectLLM[3] (EMNLP2023), which all used AUROC as their primary metric. Following the reviewer's constructive suggestions, we also supplement our results with TPR and Accuracy values (at 1% FPR) for selected datasets.
>
>     - TPR at 1% FPR（White-box/Black-box）
>         | **Datasets/Methods** | **Likelihood** | **LogRank** | **DetectLRR** | **Lastde** | **Fast-DetectGPT** | **Lastde++** |
>         | --- | --- | --- | --- | --- | --- | --- |
>         | WritingPrompts(Llama2-13b) | 11.33/1.33 | 18.67/2.00 | 40.00/19.33 | 64.67/30.00 | 63.33/10.67 | **84.67**/**50.00** |
>         | WritingPrompts(OPT-13b) | 10.00/5.33 | 19.33/9.33 | 46.67/25.33 | 71.33/**54.67** | 84.00/20.00 | **86.67**/53.33 |
>
>     - Accuracy at 1% FPR（White-box/Black-box）
>         | **Datasets/Methods** | **Likelihood** | **LogRank** | **DetectLRR** | **Lastde** | **Fast-DetectGPT** | **Lastde++** |
>         | --- | --- | --- | --- | --- | --- | --- |
>         | WritingPrompts(Llama2-13b) | 55.17/50.17 | 58.83/50.50 | 69.50/59.17 | 81.83/64.50 | 81.17/54.83 | **91.83**/**74.50** |
>         | WritingPrompts(OPT-13b) | 54.50/52.17 | 59.17/54.17 | 72.83/62.17 | 85.17/**76.83** | 91.50/59.50 | **92.83**/76.17 |
>
>     The results show that while all detection methods perform modestly when FPR is fixed at 1%, Lastde and Lastde++ consistently outperform its baseline methods. Notably, in the black-box setting, Lastde surpasses Fast-DetectGPT's performance. Additionally, we present more metrics and results under a more relaxed setting (FPR=5%) in Tables R2-R6 (in our General Response), which consistently demonstrate that the proposed method can achieve the state-of-the-art performance.
>
> - **W2:** We thank the Reviewer for raising this excellent question. We believe Lastde is fundamentally different from SeqXGPT[4] (EMNLP2023) for the following reasons:
>
>    - **a):** We apologize for the confusion and we would like to clarify that SeqXGPT is not a time series based detection method strictly, rather, it frames sentence-level text detection as a Named Entity Recognition (NER) task. Its foundational features are **directly set as the log-probability values** of each token across multiple open-source proxy models (resulting in a matrix of dimensions: token count × number of proxy models, as detailed in Section 4.3 of their paper). Therefore, we believe this approach does not involve time series transformation and analysis methods, but rather focuses on **task definition**. In contrast, our method goes beyond simply using these log-probabilities by introducing time series analysis to extract dynamic patterns from token log-probability segments, emphasizing **time series feature extraction and analysis**. Thus, we consider our approach fundamentally different from SeqXGPT in terms of perspective and feature processing.
>
>    - **b):** Another major difference is that SeqXGPT is essentially a training-based method, while our paper primarily focuses on training-free approaches. Given our extensive cross-domain corpus experiments, it might not be fair to directly comparing SeqXGPT with the proposed method on these cross-domain datasets.

---

> ### Author Response · Authors · 2024-11-23
> **1) Response to Weakness (W) : Part (2/2)**
>
> - **W3:** We would like to clarify that most of our SOTA results were achieved under the same parameter setting, and we only set different default values for Lastde and its variant, Lastde++. Based on our discussion and analysis on hyperparameters (Section 4.2), their selection does not affect Lastde or Lastde++'s ability to achieve SOTA performance among existing methods. Moreover, in Table R1 (in General Response), we unified the hyperparameter settings for Lastde and Lastde++, and the third-party data detection results still demonstrate our SOTA performance.
>
> - **W4:** In fact, GPT-2, Neo-2.7, and OPT-2.7 appear exclusively in the decoding strategy section of our robustness analysis (Section 4.3). For other results, we conducted experiments primarily on larger (7B or 13B) and more prominent models, generally following and covering Fast-DetectGPT's (ICLR 2024) experimental settings. For certain model series, we also included the latest versions (such as Llama3-8B). Due to limitations in computing resources, we couldn't generate new experimental data on source models exceeding 20B parameters, but we incorporated alternatives like Gemma and Phi2 to cover a broader range of LLM products from various tech companies. Given our coverage of model types (open-source, closed-source, new versions, old versions), which is more comprehensive than many existing studies, we believe our results are more compelling now. Here are the parameter counts for some of our source models (>7B):
>
>     | models | OPT-13 | Llama-13 | Llama2-13 | GPT-4-Turbo | GPT-4o | Claude-3-haiku |
>     | --- | --- | --- | --- | --- | --- | --- |
>     | parameters | 13B | 13B | 13B | >>7B | >>7B | >>7B |
>
> Please also refer to Table R1 (in General Response) for experimental results covering more models with ≥20B parameters.
>
> [1] Fast-DetectGPT: Efficient Zero-Shot Detection of Machine-Generated Text via Conditional Probability Curvature
>
> [2]DetectGPT: Zero-Shot Machine-Generated Text Detection using Probability Curvature
>
> [3]DetectLLM: Leveraging Log Rank Information for Zero-Shot Detection of Machine-Generated Text
>
> [4]SeqXGPT: Sentence-Level AI-Generated Text Detection

---

> ### Author Response · Authors · 2024-11-23
> **2) Response to Questions (Q)**
>
> - **Q1:** Thank you for this intriguing suggestion. Based on our understanding of this question, Lastde (ours), likelihood, rank, and lrr are all variants of f(.), and they are all training-free. We believe more training-free f() variants will be developed for LLM-generated text detection in the future. Regarding your question about "learning an optimal f(.) through a network," we believe this is entirely feasible. However, since our paper primarily focused on training-free methods, we did not extensively explored this direction, and we believe it could be a promising avenue for our future research.
>
> - **Q2:**  We agree that text generated by instruction/chat source models is generally more challenging to detect. In Appendix B.2 Table 5, we list the Hugging Face indices for all models, most of which are base models (except for closed-source models), consistent with Fast-DetectGPT's approach. In Table R1 (in General Response), we conducted detection on more chat/instruct source models, and the results demonstrate that our method still achieves the best performance.
>
> - **Q3:** Our method does not involve any assumptions about "prompts or questions used to generate AI text," therefore we do not need to know such information during detection. Indeed, all experimental results related to chat/instruct source models presented in the main paper or Table R1 (in General Response) were conducted without any prompts or questions.

---

> ### Comment · Reviewer_SbPp · 2024-11-26
> **I have raised my score**
>
> please add those to the final version

---

> > ### Author Response · Authors · 2024-11-28
> > **Thank you for raising the score, and the paper has been revised accordingly.**
> >
> > Thank you a lot for recognizing the contribution of our work and raising the score! In the revised paper, we have included these results and discussions from rebuttal accordingly.
> >
> > Please check the summary of our revision:
> >
> > - **Table 9-10 (Appendix D, Page 20-21):** In Table 9 of Appendix D, we have added a comparison of detection results on a third-party dataset between two strong baselines, GPTZero and Binoculars, and other baselines. Table 10 in Appendix D presents the results of integrating existing baselines with the lightweight TOCSIN module. All integrations and related discussions have been highlighted in red for clarity.
> > - **Table 16 (Appendix F.4, Page 25-26):** We present the detection results of our method compared with existing baselines on LLM-generated texts that mimic human writing styles in Table 16 of Appendix F.4. We have highlighted the results and corresponding analysis in red for clarity.
> > - **Table 17-19 (Appendix F.5, Page 26):** In Tables 17-19 of Appendix F.5, we provide a comparative evaluation of several efficient detectors on LLM-LLM mixture texts. These three tables respectively show AUROC, TPR (at 5% FPR), and Accuracy (at 5% FPR). All results and discussions have been highlighted in red for clarity.
> > - **Table 20 (Appendix G, Page 27):** We present the detection results using the decision boundary of a univariate logistic regression model as the scoring threshold in Table 20 of Appendix G. We also provide an improved threshold determination method, with all discussions and improvements highlighted in red.
> >
> > Thanks!
> >
> > The Authors

---

### Author Response · Authors · 2024-11-23
**General Responses : Part II (2/2)**

**3) Detecting LLM/human mixted texts:** We econsidered three different types of mixed texts, i.e., a) LLM-LLM mixture text, b) Human-LLM co-authored text, and c) LLM-revised text, and reported our results below.

**a) LLM-LLM mixture text** Using WritingPrompts as an example, we constructed datasets with text generated by multiple different LLMs. Specifically:

We experimented on detecting mixed texts from two LLM models and four LLM models, respectively. In the two-model mixture experiments, we evaluated on two different mixture ratios, i.e.,  50%+50% (Llama2-13B,OPT-13B) and 80%+20% (BLOOM-7B,Falcon-7B), respectively. In the four-model mixture experimenet, we used 25% texts from each of the four LLM models (Llama2-13B,OPT-13B,BLOOM-7B,Falcon-7B). We employed a simple "truncate-concatenate" approach to generate the mixed data. For example, taking 50% of characters from the first source model's text and concatenating with 50% from the second model's text. While this may lose some contextual information, we believe it's sufficient and efficient for demonstration purposes. The experimental results are as follows:

Table R4: Experimental results on LLM-LLM mixture text (AUROC metric)
| Mixing ratio | SourceModels | Likelihood | LogRank | DetectLRR | **Lastde** | Fast-DetectGPT | **Lastde++** |
| --- | --- | --- | --- | --- | --- | --- | --- |
| 50%+50%  | (Llama2-13B,OPT-13B) | 81.48 | 84.81 | 87.99 | 91.88 | 85.21 | **92.48** |
|  | (BLOOM-7B,Falcon-7B) | 78.80 | 82.56 | 87.47 | 94.95 | 88.59 | **95.65** |
| 80%+20%  | (Llama2-13B,OPT-13B) | 80.44 | 82.97 | 85.41 | 90.69 | 86.73 | **92.64** |
|  | (BLOOM-7B,Falcon-7B) | 76.98 | 81.52 | 52.91 | 96.38 | 92.05 | **96.57** |
| 25% each | (Llama2-13B,OPT-13B,BLOOM-7B,Falcon-7B) | 82.33 | 86.34 | 88.02 | **94.44** | 86.02 | 93.32 |

Table R5: Experimental results on LLM-LLM mixture text (TPR at 5% FPR metric)
| Mixing ratio | SourceModels | Likelihood | LogRank | DetectLRR | **Lastde** | Fast-DetectGPT | **Lastde++** |
| --- | --- | --- | --- | --- | --- | --- | --- |
| 50%+50%  | (Llama2-13B,OPT-13B) | 29.33 | 36.00 | 40.00 | 65.33 | 46.67 | **68.00** |
|  | (BLOOM-7B,Falcon-7B) | 16.00 | 30.00 | 34.67 | **76.00** | 39.33 | 70.67 |
| 80%+20%  | (Llama2-13B,OPT-13B) | 24.67 | 28.67 | 38.67 | 58.67 | 41.33 | **69.33** |
|  | (BLOOM-7B,Falcon-7B) | 16.67 | 22.67 | 44.67 | 80.67 | 59.33 | **83.33** |
| 25% each | (Llama2-13B,OPT-13B,BLOOM-7B,Falcon-7B) | 22.00 | 40.67 | 46.00 | **81.33** | 40.67 | 73.33 |

Table R6: Experimental results on LLM-LLM mixture text (Accuracy at 5% FPR)
| Mixing ratio | SourceModels | Likelihood | LogRank | DetectLRR | **Lastde** | Fast-DetectGPT | **Lastde++** |
| --- | --- | --- | --- | --- | --- | --- | --- |
| 50%+50%  | (Llama2-13B,OPT-13B) | 62.17 | 65.50 | 67.50 | 80.17 | 70.83 | **81.50** |
|  | (BLOOM-7B,Falcon-7B) | 55.55 | 62.50 | 64.83 | 85.50 | 67.17 | **82.83** |
| 80%+20%  | (Llama2-13B,OPT-13B) | 59.83 | 61.83 | 66.83 | 76.83 | 68.17 | **82.17** |
|  | (BLOOM-7B,Falcon-7B) | 55.83 | 58.83 | 69.83 | 87.83 | 77.17 | **89.17** |
| 25% each | (Llama2-13B,OPT-13B,BLOOM-7B,Falcon-7B) | 58.50 | 67.83 | 70.50 | **88.17** | 67.83 | 84.17 |

 The results show that even with texts mixed from different source models and different ratios, Lastde and Lastde++ still achieve good detection performance. In all three metrics across all datasets, Lastde or Lastde++ reach SOTA, especially with TPR (at 5% FPR) exceeding Fast-DetectGPT, the prior SOTA, by at least 20%.

**b) Human-LLM co-authored text**：Many studies (Fast-DetectGPT [1] , DNA-GPT [2] (ICLR2024), DetectGPT [3] (ICML2023)) use the T5 model to simulate human modifications of text. Since the T5_Paraphraser used in the main text only attacked LLM-generated text, the resulting text can be considered as co-authored by human (T5 mimic) and LLM. We provided the complete detection results in Table 3 on page 9 of the main text, and here is a summary of the results:
| **Methods** | Xsum(Paraphrased) | WritingPrompts(Paraphrased) | Reddit(Paraphrased) |
| --- | --- | --- | --- |
| Fast-DetectGPT | 67.17 | 83.82 | 83.04 |
| Lastde++ | **73.43** | **88.54** | **87.90** |

Compared to Fast-DetectGPT, Lastde++ shows higher detection performance on Human-LLM co-authored text (post-attack text), indicating its greater robustness.

**c) LLM-revised text:**  For LLM-revised text experiment, we used a powerful paraphraser to attack both human-written text and LLM-generated text simultaneously, and the results are shown in Table R3. We believe that the post-attack text falls under one scenario of LLM-revised text. Table R3 shows that Lastde++ is more adaptable to LLM-revised text compared to Fast-DetectGPT, indicating its stronger robustness.

[1] Efficient Zero-Shot Detection of Machine-Generated Text via Conditional Probability Curvature.

[2] Divergent N-Gram Analysis for Training-Free Detection of GPT-Generated Text.

[3] Zero-Shot Machine-Generated Text Detection using Probability Curvature.

---

### Author Response · Authors · 2024-11-23
**General Responses : Part II (1/2)**

**Concern II** More experiments on evaluating the robustness of our method.

**Response to Concern II** In this part, we address another three research questions, including: capability in detecting LLM-generated texts that mimic human writing styles, showing robustness against more powerful paraphrasers, superior performance in detecting LLM-LLM mixed text, Human-LLM co-authored text, and LLM-revised text.

**1) Detecting human-style texts:** Using the WritingPrompts dataset as an example, we created two new datasets using gpt-4-turbo-04-09 and gpt4o-08-06. Unlike before, we designed the prompt as: "Please mimic the human writer an article with about 200 words starting exactly with: <prefix>", explicitly requiring human-style mimicry. We used the settings from Appendix D Table 8 of our paper, with Lastde++ directly using expstd. The following Table R2 show results for three metrics:

Table R2: The average of WritingPrompts (gpt-4-turbo-0409) and WritingPrompts (gpt4o-08-06)
|                        | Likelihood | LogRank | DetectLRR | **Lastde** | Fast-DetectGPT | **Lastde++** |
| ---------------------- | ---------- | ------- | --------- | ---------- | -------------- | ------------ |
| **AUROC**              | 87.37      | 85.60   | 74.94     | 87.68      | **96.49**      | **96.49**    |
| **TPR at 5% FPR**      | 30.00      | 30.67   | 24.34     | 29.33      | 82.67          | **85.36**    |
| **Accuracy at 5% FPR** | 62.50      | 62.83   | 59.67     | 63.34      | 88.84          | **90.17**    |

For both human-style mimicking datasets, Lastde++ shows comparable or better results across all three metrics compared to Fast-DetectGPT [1] (ICLR2024), demonstrating its effectiveness in detecting LLM text that mimics human writing style.

**2) More powerful paraphraser:**  The T5_Paraphraser used in our paper is an 800M parameter model (consistent with Fast-DetectGPT). To further demonstrate our method's robustness, we explored the most downloaded paraphraser on HuggingFace specifically designed for text attacks. Since Dipper's model exceeds 50G, we couldn't run it on our hardware. Instead, we chose another popular ~3G model: t5-large-paraphraser-diverse-high-quality (https://huggingface.co/ramsrigouthamg/t5-large-paraphraser-diverse-high-quality). Below are the detection results (AUROC) on WritingPrompts (Llama2-13B) and WritingPrompts (OPT-13B) before and after attacking with the new paraphraser:

Table R3: Experimental results on more powerful paraphrasing attack
| **Scenarios** | **Methods/Datasets** | **Llama2-13B(Original)** | **Llama2-13B(Paraphrased)** | **OPT-13B(Original)** | **OPT-13B(Paraphrased)** |
| --- | --- | --- | --- | --- | --- |
| **White-box** | Fast-DetectGPT | 98.04 | 88.59(-9.45) | **99.40** | 89.42(-9.98) |
|  | Lastde++ | **99.41** | **93.41(-6.00)** | **99.40** | **90.81(-8.59)** |
| **Black-box** | Fast-DetectGPT | 86.44 | 78.79(-7.65) | 89.64 | 84.00(-5.64) |
|  | Lastde++ | **93.05** | **85.86(-7.19)** | **95.25** | **90.86(-4.39)** |

Analysis: Even under attacks from a more powerful paraphraser, Lastde++ shows smaller performance degradation compared to Fast-DetectGPT in both white-box and black-box scenarios, indicating its superior robustness against paraphrasing attacks.

---

### Author Response · Authors · 2024-11-23
**General Responses : Part I**

We would like to sincerely thank all reviewers for their time and constructive comments. After carefully considering all comments, we have categorized common suggestions into two main groups, and addressed all concerns accordingly in a point-by-point manner. We summarize our response and revisions below.

**Concern I** Comparisons with stronger baselines and source models, and exploring the possibility with some recent plug-and-play detection methods.

**Response to Concern I**. In this part, we address four research questions, including: verifying the superior performance of our method on more datasets with stronger source models (e.g. chat/instruct model), invloving more powerful baselines (e.g. GPTZero), improved performance by integrating with learning-based detectors, and hyperparameter tuning. Below are our detailed responses.

**1) Including more datasets:** We conducted experiments on multiple datasets recently released by ReMoDetect [1] (NeurIPS2024) and Fast-DetectGPT [2] (ICLR2024). These datasets cover four domains: Xsum, SQuAD, WritingPrompts, PubMedQA, and six chat/instruct type or large-parameter (≥20B) source models: NeoX-20B, Llama3-70B, GPT-3.5-Turbo, GPT-4, GPT-4-Turbo, Claude3. The dataset links are as follows (for Fast-DetectGPT, we only use data generated by NeoX-20B):
- ReMoDetect (Llama3-70B、GPT-3.5-Turbo、GPT-4、GPT-4-Turbo、Claude3) : https://github.com/hyunseoklee-ai/ReMoDetect/tree/main/exp/data
-  Fast-DetectGPT(NeoX-20B) : https://github.com/baoguangsheng/fast-detect-gpt/tree/main/exp_main/data

**2) Including more baselines:** To make the experiments more complelling, we have added another two strong baselines: Binoculars [3] (ICML2024), GPTZero (commercial, 2024-11-11 base version). For Binoculars, we adopted its default settings from the original paper. For GPTZero, we first replicated the detection results provided in the ReMoDetect paper [1] (NeurIPS2024), then performed detection experiments on the Fast-DetectGPT (NeoX-20B) related datasets.

**3) Combining with plug-and-play detector:** We explored the feasibility of integrating our method with recent plug-and-play detectors (plugins), e.g. TOCSIN [4] (EMNLP2024). Specifically, in the rebuttal, we combined TOCSIN with Lastde and Lastde++ (the same approach was applied to other baselines), and reported the detection results (AUROC values) on the aforementioned multiple datasets.

**4) Unified hyperparameters:** We unified the hyperparameters of Lastde and Lastde++ to $s=3,\varepsilon=10,\tau^{\prime}=5$, as the parameters used here (not the best parameters in our datasets).  For other baseline methods, we kept their sampling and scoring models as GPT-J.

**5) Our new results:** We conducted experiments following the above settings, and reported our new results in Table R1, which includes before and after integrating with TOCSIN. Table R1 shows the average AUROC across all datasets, and the complete results have been added to Appendix of our revised paper and highlighted in red font.


Table R1: Comparisons with more datasets and more baselines.
|                | Likelihood | LogRank | DetectLRR | Fast-DetectGPT | Binoculars | GPTZero | **Lastde** | **Lastde++** |
| -------------- | ---------- | ------- | --------- | -------------- | ---------- | ------- | ---------- | ------------ |
| without TOCSIN | 87.0       | 87.2   | 82.2      | 93.7           | 87.5      | 88.6    | 89.6       | **94.2**     |
| with TOCSIN    | 92.4       | 92.4    | 85.5      | 94.3           | 87.5       | 88.6    | 93.3   | **95.3**         |
| improvement    | 5.4       | 5.2    | 3.3      | 0.6            | -          | -       | 3.7       | 1.1          |

Analysis: Table R1 shows that our method can still achieve SOTA even if we did not run Lastde++ (Distribution-based) under the optimal hyperparameter setting. This conclusion holds true even when compared to the two newly added baselines: Binoculars, GPTZero. Meanwhile, Lastde also performed best among Sample-based methods. Finally, all detection methods showed performance improvements when combined with TOCSIN, a lightweight plug-and-play detection module, with our method showing more significant gains (compared with Fast-DetegtGPT). With more datasets and more powerful baselines, we believe that Lastde and Lastde++ are highly effective detectors.

[1] ReMoDetect: Reward Models Recognize Aligned LLM’s Generations, NeurIPS 2024

[2] Fast-DetectGPT: Efficient Zero-Shot Detection of Machine-Generated Text via Conditional Probability Curvature, ICLR 2024

[3] Binoculars: Spotting LLMs With Binoculars: Zero-Shot Detection of Machine-Generated Text, ICML 2024

[4] TOCSIN: Zero-Shot Detection of LLM-Generated Text using Token Cohesiveness, EMNLP 2024

---

### Meta-Review · Area_Chair_zrqo · 2024-12-20

**Metareview:**

- Scientific Claims and Findings:
    - The paper introduces a training-free approach for detecting text generated by LLMs. It leverages time series analysis to mine token probability sequences, effectively combining local and global statistics for improved detection accuracy.

- Strengths:
   - The proposed method is new and sound.
   - It is effective across cross-domain and cross-model scenarios without requiring retraining, and it is robust against paraphrasing attacks and various textual manipulations.

- Weaknesses:
    - The method may be somewhat complex due to the many hyperparameters that need tuning.

- Most Important Reasons for Decision:
     - Based on the identified strengths.

**Additional Comments On Reviewer Discussion:**

In their rebuttal, the authors clarified the distinctions between their proposed method and some existing approaches like SeqXGPT, effectively addressing one of the main concerns raised by Reviewer SbPp. They also conducted additional experiments by incorporating more datasets, comparing against more baselines, and exploring diverse configurations and attack scenarios to comprehensively address the reviewers' concerns.

Following the rebuttal, Reviewer SbPp increased their rating from 5 to 6, while the other reviewers maintained their ratings of 6. All reviewers are inclined to accept the paper.

---

### Decision · Program_Chairs · 2025-01-22

Accept (Poster)